# Claustrum neurons projecting to the anterior cingulate restrict engagement during sleep and behavior

Gal Atlan[1], Noa Matosevich[2,3,9], Noa Peretz-Rivlin[1,9], Idit Marsh-Yvgi[4,9], Noam Zelinger[2,3], Eden Chen[1], Timna Kleinman[1], Noa Bleistein[1,4], Efrat Sheinbach[1,4], Maya Groysman[1], Yuval Nir [2,3,5,6,7] & Ami Citri [1,4,8] ✉

The claustrum has been linked to attention and sleep. We hypothesized that this reflects a shared function, determining responsiveness to stimuli, which spans the axis of engagement. To test this hypothesis, we recorded claustrum population dynamics from male mice during both sleep and an attentional task ('ENGAGE'). Heightened activity in claustrum neurons projecting to the anterior cingulate cortex (ACCp) corresponded to reduced sensory responsiveness during sleep. Similarly, in the ENGAGE task, heightened ACCp activity correlated with disengagement and behavioral lapses, while low ACCp activity correlated with hyper-engagement and impulsive errors. Chemogenetic elevation of ACCp activity reduced both awakenings during sleep and impulsive errors in the ENGAGE task. Furthermore, mice employing an exploration strategy in the task showed a stronger correlation between ACCp activity and performance compared to mice employing an exploitation strategy which reduced task complexity. Our results implicate ACCp claustrum neurons in restricting engagement during sleep and goal-directed behavior.

Engagement describes the degree to which a subject is likely to interact with external sensory stimuli[1–3]. Engagement exists along a continuum; from low levels of engagement in sleep, environmental detachment, and suppression of sensory-evoked responses, through varying levels during wakefulness[4,5]. During wakefulness, states of disengagement can appear as sleep-like states and are accompanied by behavioral lapses[6–8], while states of hyper-engagement risk the execution of impulsive responses to irrelevant stimuli[2,9,10]. The neuronal substrate of engagement is rarely studied as a continuum, but hypo-engaged and hyper-engaged behavioral states have been associated with distinct cortical activity patterns[5,6,8,11–13].

The claustrum, a subcortical structure with extensive cortical connectivity, has been recently implicated in the modulation of sleep[14–16], particularly non rapid eye movement (NREM) sleep[17–19]. In parallel, the claustrum has been functionally associated with multiple aspects of cognition and behavior, such as attention, salience, impulse control, and action planning[20–33]. Human lesion studies likewise implicate the claustrum in perception, salience, as well as in sleep[22]. Given its apparent role in sleep on one hand, and task responsivity on the other, we set out to test the hypothesis that the claustrum can act to restrict engagement and responsivity to sensory stimuli across the spectrum of behavioral states, in sleeping as well as in behaving mice.

[1]The Edmond and Lily Safra Center for Brain Sciences, The Hebrew University of Jerusalem; Edmond J. Safra Campus, Givat Ram, Jerusalem, Israel. [2]Department of Physiology & Pharmacology, Faculty of Medical and Health Sciences, Tel Aviv University, Tel Aviv, Israel. [3]Sagol School of Neuroscience, Tel Aviv University, Tel Aviv, Israel. [4]The Alexander Silberman Institute of Life Science, Faculty of Science, The Hebrew University of Jerusalem; Edmond J. Safra Campus, Givat Ram, Jerusalem, Israel. [5]Department of Biomedical Engineering, Faculty of Engineering, Tel Aviv University, Tel Aviv, Israel. [6]The Sieratzki-Sagol Center for Sleep Medicine, Tel Aviv Sourasky Medical Center, Tel Aviv, Israel. [7]Sagol Brain Institute, Tel Aviv Sourasky Medical Center, Tel Aviv, Israel. [8]Program in Child and Brain Development, Canadian Institute for Advanced Research; MaRS Centre, Toronto, ON, Canada. [9]These authors contributed equally: Noa Matosevich, Noa Peretz-Rivlin, Idit Marsh-Yvgi. ✉e-mail: Ami.Citri@mail.huji.ac.il

The claustrum consists of functionally segregated networks, necessitating the use of projection-based approaches for investigating the roles of claustral circuits[34–38]. The anterior cingulate cortex (ACC) is a central target of claustrum projections[17,24,30,32–34,36–52], and stimulation of claustral neurons projecting to the cingulate cortex (ACCp neurons) has been shown to elicit a prominent effect on cortical activity[17,33]. Furthermore, the ACC is implicated in impulse control and the modulation of behavioral sensory responsiveness[53,54]. As such, ACCp neurons in the claustrum are a particularly promising candidate population for mediating executive control over engagement.

In this study we utilized fiber photometry and chemogenetic modulation to selectively record and manipulate the activity of ACCp neurons during NREM sleep, wakefulness, and a novel psychometric task designed to assess engagement. Through the analysis of the behavioral strategies of individual mice, we identified a correlation between claustrum dynamics and behavioral outcomes, that depends on the implemented strategy. Our results collectively highlight a pivotal role of ACCp claustrum neurons in mediating engagement and sensory responsiveness, extending from sleep to task performance.

## Results

### Elevated ACCp claustrum activity promotes unresponsive sleep

Recent studies have associated claustrum activity with sleep in mice and reptiles[17,18]. Since a key property of sleep is reduced behavioral responsiveness to external sensory stimuli, we set out to test whether claustrum activity modulates the probability of sensory-evoked awakenings. To this end, we recorded the activity of claustrum neurons that project to the anterior cingulate cortex (ACCp) using calcium fiber photometry in unrestrained mice, while simultaneously performing polysomnography (video, EEG – electroencephalography and EMG – electromyography) to determine their sleep state (Figs. 1A, B and S1A, B. Fig. S1C, D provides a reference for the stability of the photometry recordings over 12-h sessions). Mice displayed typical sleep patterns over the course of the recording sessions, spending $54.0 \pm 7.1\%$ of the time in NREM; $34.4 \pm 5.8\%$ awake; and $8.1 \pm 1.0\%$ in REM sleep (Fig. S1A). In line with previous findings, we observed ACCp activity to be elevated during NREM sleep in comparison to wakefulness (Figs. 1C and S1C)[17,18]. To go beyond spontaneous changes in sleep states, we investigated the relation of ACCp activity to sensory responsivity, by delivering intermittent sensory stimuli during NREM sleep and determining whether the mouse awakened or maintained its sleep (see "Methods" section). In this tone-evoked arousal experiment we found that ACCp activity preceding the stimulus was significantly higher in 'sleep-maintained' trials in comparison to 'awakening' trials (Fig. 1D, E; "Methods" section[55]). This association was not observed when analyzing activity of a population of claustrum neurons captured by their projection to a different cortical target, the orbitofrontal cortex (OFCp neurons) in another set of animals, illustrating that this is not a global effect, but rather one that maintains specificity to ACCp activity (Fig. S1E–G; for a comparison between ACCp and OFCp populations, see Fig. S2). Thus, our results establish a specific correlation between elevated ACCp claustrum activity and unresponsive sleep.

To address the causal role of ACCp activity in promoting unresponsive sleep, we expressed the excitatory DREADD hM3Dq in ACCp neurons (Figs. 1F and S1H), and examined the impact of elevated ACCp activity on sleep and sensory responsiveness. We conducted blind experiments administering either clozapine n-oxide (CNO, 10 mg/kg, i.p.) or saline (on different days, with order counter-balanced), separately in hM3Dq mice ($n = 6$) or control mice ($n = 7$) injected with virus expressing fluorophore-only. DREADD experiments were conducted during undisturbed sleep and during tone-evoked arousal experiments on separate days (Fig. 1G, "Methods" section). Figure 1H presents hypnograms (time-course of vigilance states) in a representative mouse showing the effects of ACCp claustrum activation on spontaneous sleep. We observed a consistent and robust increase in the prevalence of NREM sleep in the hours following activation of ACCp claustrum neurons (Methods), increasing from $46.3 \pm 7.2\%$ following saline injections to $65.6 \pm 5.4\%$ following CNO injections (Fig. 1I). Control mice did not exhibit such a significant difference (Fig. S1I), indicating that CNO alone could not account for the observed effects. Next, to directly address the effects on unresponsiveness during sleep, we tested the effect of activation of ACCp claustrum neurons in the tone-evoked arousal experiment. Examples of awakening and maintained trials are presented in Fig. 1J, and representative effects of CNO or saline injections in one mouse are displayed in Fig. 1K. Across the entire dataset, we observed a significant decrease in sound-evoked awakening probability in the hours following activation of ACCp claustrum neurons (Methods), from $11.2 \pm 4.9\%$ following saline injections to $5.2 \pm 2.2\%$ following CNO injections (Fig. 1L). Again, control mice did not exhibit such a significant difference (Fig. S1J). Altogether, both correlative and causal manipulation data establish association between elevated ACCp claustrum activity and unresponsive NREM sleep.

### Pre-trial activity of ACCp neurons restricts engagement in the ENGAGE task

Based on our observation that elevated ACCp activity promotes unresponsiveness during sleep, we hypothesized that a similar logic may apply to engagement during behavior. If so, we would expect to observe reduced ACCp activity when mice are hyper-engaged and elevated ACCp activity when mice are disengaged. To test these predictions, we developed 'ENGAGE', a task aimed at challenging multiple aspects of task engagement, including response inhibition, selective attention, and sustained attention (Fig. 2A)[56]. Full details of the parameters of the ENGAGE task are described in the methods section.

Briefly, each trial in the task is initiated with the presentation of a broad-band-noise (BBN) 'No Go' cue, followed by a pure-tone 'Go' cue (presented at one of four intensities). The 'No Go' and 'Go' cues are separated by a delay period, randomly lasting for $0.5$–$3$ s. Mice must refrain from licking throughout the presentation of the 'No Go' cue and the delay period, until the 'Go' cue is presented. In addition, a tone cloud stimulus is presented throughout 50% of the trials, masking the 'Go' cue. To maintain participation under these challenging conditions[57–59], a visual aid (an LED flash that turned on concurrently with the auditory 'Go' cue) was provided in 25% of the trials (Fig. 2B). To succeed in the task, mice must generate appropriately gated responses to the 'Go' cue ('Hits') in order to obtain reward. Doing so requires a balance between response inhibition, limiting premature licking during the window following the 'No Go' BBN and prior to the 'Go' cue (defined as 'Impulsive errors' and terminating the trial); and behavioral lapses, which lead to missed 'Go' cues ('Misses')[60].

To account for the possibility that the claustrum is involved differentially as a function of cognitive demand, the 'Go' cue was presented at different intensities, challenging selective attention and cue detection[57,59]. Additionally, we introduced tone cloud masking, which further challenged selective attention and cue detection, while also enhancing the difficulty of response inhibition[57–59,61]. Sustained attention was challenged at different time scales, by the randomized delay to cue presentation, long inter-trial intervals (20 seconds), and the prolonged session structure (typically 360 trials over the course of 2 h)[62,63].

Performance in the task (in sum ~100,000 trials from 24 mice, see Supplementary Table 1) was, on average, impacted by all trial variables: hit rates (i.e. success rate in trials in which the 'Go' cue was presented) improved psychometrically with cue intensity, while the visual aid was most beneficial at low cue intensities. The tone cloud induced a general increase in impulsive error rates, as well as a reduction in cue detection, evident as reduced hit rates in trials with intermediate cue intensities (Fig. 2C).

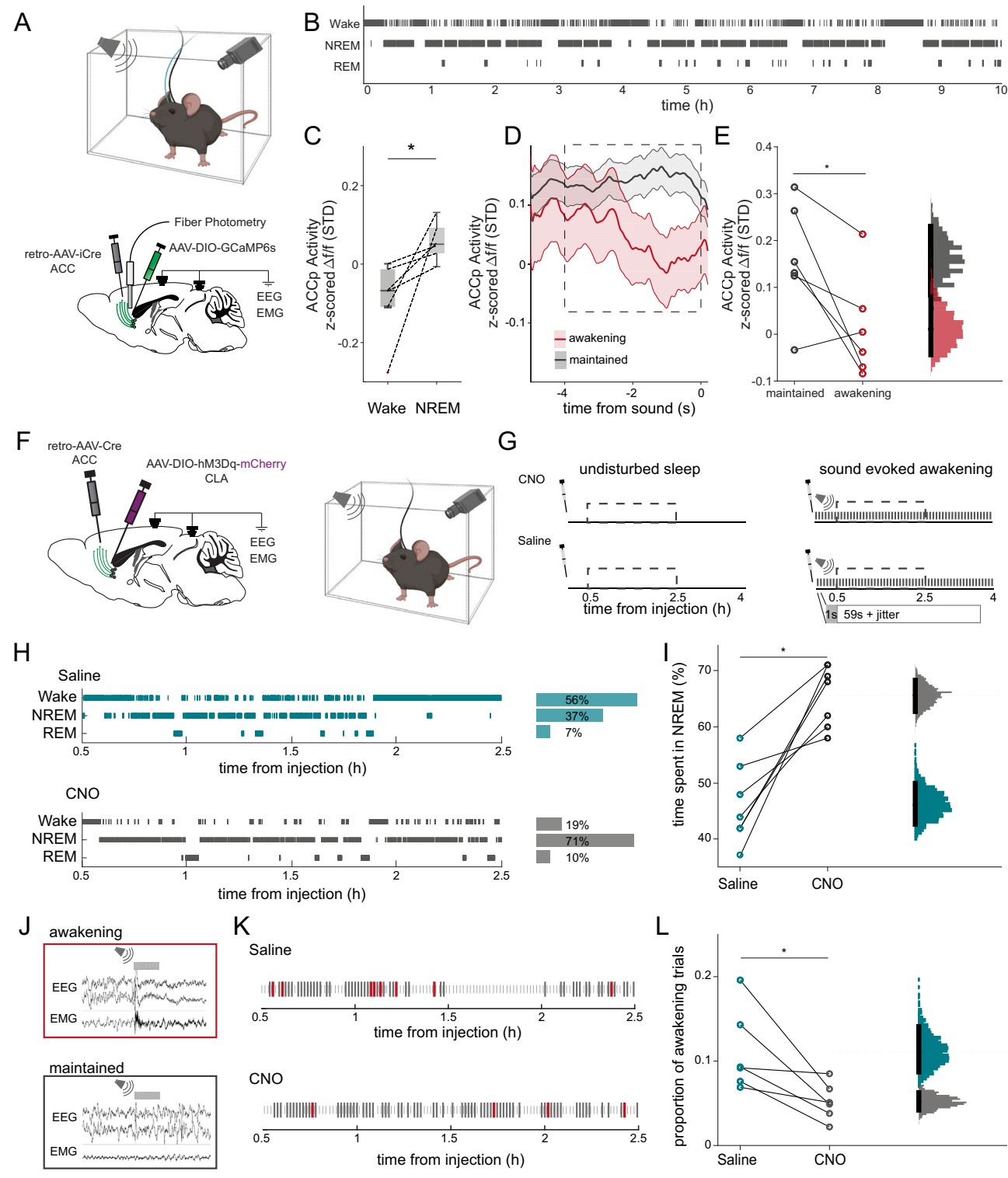

We recorded fiber photometry from ACCp neurons as trained mice performed the ENGAGE task (Fig. 2D). Addressing the representation of task-associated events within the ACCp signal, we observed lick-associated ACCp transients following both impulsive errors as well as rewarded 'Hit' trials (Fig. 2E, F). Furthermore, we observed evoked transients following the 'Go' cue which, on average, scaled with their salience (Fig. 2G). To quantify the relative representation of the different behavioral events within the photometry signal, we applied a generalized linear encoding model[64,65]. This analysis showed that ACCp neurons demonstrated significant evoked activity corresponding to spontaneous locomotion

(Fig. S3D), task-related licking (Fig. S3E), and sensory stimuli (Fig. S3F).

Based on our finding that elevated pre-stimulus ACCp activity was associated with the maintenance of unresponsive sleep, we hypothesized that pre-trial ACCp activity could relate to task disengagement. Indeed, on average, pre-trial ACCp activity showed a clear separation based on the outcome of the subsequent trial (Fig. 2H, I). Compared to baseline activity before hits, pre-trial activity was elevated before miss trials and reduced prior to impulsive errors (Fig. 2H, I). In fact, binning pre-trial ACCp activity into quantiles demonstrated an inverse correlation of pre-trial activity with impulsive errors: lower ACCp activity

**Fig. 1 | ACCp activity promotes sensory unresponsiveness during NREM sleep.**
**A** Top: experimental setup. Bottom: simultaneous EEG, neck EMG, and ACCp fiber photometry. **B** Representative hypnogram (black ticks, 4 s data epochs). **C** Average ACCp activity by sleep states. Dots represent individual mice. Boxes represent the median, and 25th and 75th percentiles, with whiskers extending to non-outlier extremes ($n = 6$, $t_{(5)} = -2.72$, $p = 0.041$, paired $t$-test). **D** ACCp photometry time-course preceding a tone (presented at time 0) in arousal experiments during NREM sleep. Data represented as mean ± s.e.m. Dashed square represents the time window analyzed. **E** Difference in ACCp activity preceding the tone. Each dot represents average activity from a single animal ($n = 6$, $t_{(5)} = 2.72$, $p = 0.042$, paired $t$-test). Gray: maintained sleep; Red: awakening. Histograms depict the bootstrapped distribution of means with 95% confidence intervals. **F** Approach for hM3Dq DREADD experiments (left) and setup (right). **G** Chemogenetic experimental timeline. Mice were injected with either CNO (top) or vehicle (bottom), and participated in an undisturbed sleep recording (left), and in an auditory threshold experiment (right).

Analysis was performed on a 2 hr window (dashed box). **H** Hypnogram from representative mouse during undisturbed sleep after CNO injection (bottom, gray) and after vehicle injection (top, teal). Bar graphs to the right show percentage of time spent in each state during recording. **I** Percent time spent in NREM sleep. Each dot represents a mouse ($n = 6$, $t_{(5)} = 5.02$, $p = 0.0024$, paired $t$-test). Gray: CNO; teal: vehicle. Histograms depict bootstrapped distributions as in panel E. **J** Auditory arousal threshold experiments during NREM sleep. Representative traces of frontal EEG (top), parietal EEG (middle), and EMG (bottom) in trials associated with awakening (top, red frame) vs. maintained NREM sleep (bottom, gray frame). Gray horizontal rectangle above represents 1 s broadband noise sound stimulus. **K** Trials during NREM sleep after CNO injection (top), and vehicle injection (bottom) from a single mouse. black ticks, maintained trials; red ticks, awakenings. Smaller gray ticks, wake or REM sleep. **L** Proportion of awakenings. Gray: CNO; teal: vehicle ($n = 6$, $t_{(5)} = -3.17$, $p = 0.024$, paired $t$-test). Histograms depict bootstrapped distributions as in panel **E**.

corresponded to a higher probability that the trial would result in an impulsive error (Fig. 2J). The opposite correlation was observed between pre-trial ACCp activity and misses: higher ACCp activity corresponded to a higher probability of task omission (Fig. 2K). Importantly, while the profile of motor- and sensory-evoked activity was similar between ACCp and OFCp neurons (Fig. S4A–E), we observed no significant relationship between pretrial OFCp activity and trial outcome, illustrating the specificity of the relation of ACCp neurons and trial outcome (Fig. S4F–I). These results support the hypothesis that the activity of the ACCp subpopulation of claustrum neurons reflects the engagement level of mice, consistent with our observations regarding ACCp activity in sleep. ACCp activity is low when mice are hyper-engaged and prone to impulsively react to 'No Go' stimuli. ACCp activity is elevated when mice are hypo-engaged and prone to omit responses to the Go stimulus.

## Chemogenetic elevation of ACCp activity reduces impulsivity in the ENGAGE task

To test whether elevating ACCp activity could induce a causal shift in task engagement, similar to the effect we observed during sleep, we co-expressed the excitatory DREADD hM3Dq together with GCaMP6s in ACCp neurons of a new cohort of mice ($n = 5$ hM3Dq and $n = 5$ control mice; see Supplementary Table 1 and Fig. 3A). This experimental setup enabled a direct validation of the effects of chemogenetic manipulation on ACCp activity, which was reliably increased following CNO administration (10 mg/kg, i.p) in experimental mice, but not in control mice (Figs. 3B, C and S5A). These mice were then trained in the ENGAGE task, as described above, and habituated to intraperitoneal (i.p) injections of saline prior to head-restrained behavioral sessions. Finally, the mice were then tested on interleaved days of either saline or CNO administration, repeated three times for each animal. In hM3Dq-expressing mice, CNO significantly and reversibly reduced impulsive error rates (Fig. 3D). Hit rates remained overall unaffected (Figs. 3E and S5B), while on average, miss rates increased following CNO administration, but this increase was not statistically significant when accounting for individual mice (Figs. 3E and S5C). Furthermore, the mice's overall success rate (Fig. S5C) and reaction time in hit trials (Fig. S5D) remained unaffected, ruling out the possibility that these mice demonstrate motor impairments. The effect of increased ACCp activity on the engagement of mice was reflected in a change in the distribution of trial outcomes throughout a session. Under CNO, impulsive errors in the first half of a session were largely replaced by hits, while in the second half of the session miss trials became more common (Fig. 3F). The significant reduction in impulsive error rate contrasts with the absence of any impact of increased ACCp activity on the modulation of performance by the tone-cloud or the visual aid (Fig. 3G). As an additional measure of decreased engagement, we observed an increase in streaks of consecutive missed trials upon administration of CNO (Fig. S5E). Importantly, we observed no

effect of CNO administration on the impulse or miss rates in mice expressing only GcAMP6s, controlling for non-specific effects of CNO on performance in the ENGAGE task (Fig. S5F, G). Overall, these findings demonstrate a causal role for elevated ACCp activity in restricting task engagement.

## ACCp activity is associated with an exploratory behavioral strategy

We next investigated whether the degree to which claustrum activity relates to task engagement and trial outcome may vary as a function of the strategy employed to resolve the task.

In humans, in other primates, and also in rodents, individual strategies are commonly assessed along the exploration/exploitation axis[66–70]. In the ENGAGE task, mice might prioritize high signal-to-noise trials (e.g., trials with prominent cues) to develop a simplified cue-response strategy (a form of exploitation) and avoid impulsive errors. The tradeoff of employing such a strategy lies in the cost associated with forfeiting attempts to respond to low-prominence cues, resulting in a notable proportion of missed trials. In contrast, mice could attempt to respond to all cue types, increasing the range of experimental parameters with which they engage (a form of exploration), at the risk of committing impulse errors[66,67]. As done for the hM3Dq-expressing mice (Fig. 3G), we plotted the performance of the entire cohort of recorded mice ($n = 25$) as a function of their impulse errors (hyper-engagement) vs. the detrimental impact of the tone-cloud on their hit rates (a proxy for sensory selectivity, quantified as a modulation index). This analysis segregated mice into two distinct behavioral groups (Fig. 4A). The first group ('exploiters', $n = 9$) comprised of mice exhibiting low impulse error rates and low cloud modulation. Exploiter mice also benefitted the most from the presence of the visual aid, similarly quantified as a high visual modulation index (Fig. 4A). The second group of mice ('explorers', $n = 16$) performed more impulse errors, were more susceptible to the presence of the tone cloud, and benefitted less from the visual aid. Plotting the average psychometric curves for these groups of mice further illustrated their differential strategies (Fig. 4B). Exploiter mice employed a simplified strategy, forgoing difficult trials in favor of successfully engaging in trials in which either a visual aid or a prominent auditory cue were presented. In contrast, explorer mice showed more uniform participation across all trial types and were susceptible to the presence of the tone cloud 'mask'. This was evident as an increase in impulsive errors in the presence of the tone-cloud mask in explorer mice (Fig. 4C). Furthermore, the impulsive errors of explorers were tightly coupled to the 'No Go' BBN presented on trial onset, while the fewer impulsive errors performed by exploiters were more uniformly distributed throughout the trial, indicating that explorers experienced more difficulty in suppressing their licks in response to the 'No Go' BBN and the tone-cloud (Fig. 4D). Importantly, while exploiter mice responded to a smaller proportion of trials compared to explorers, their overall success rate

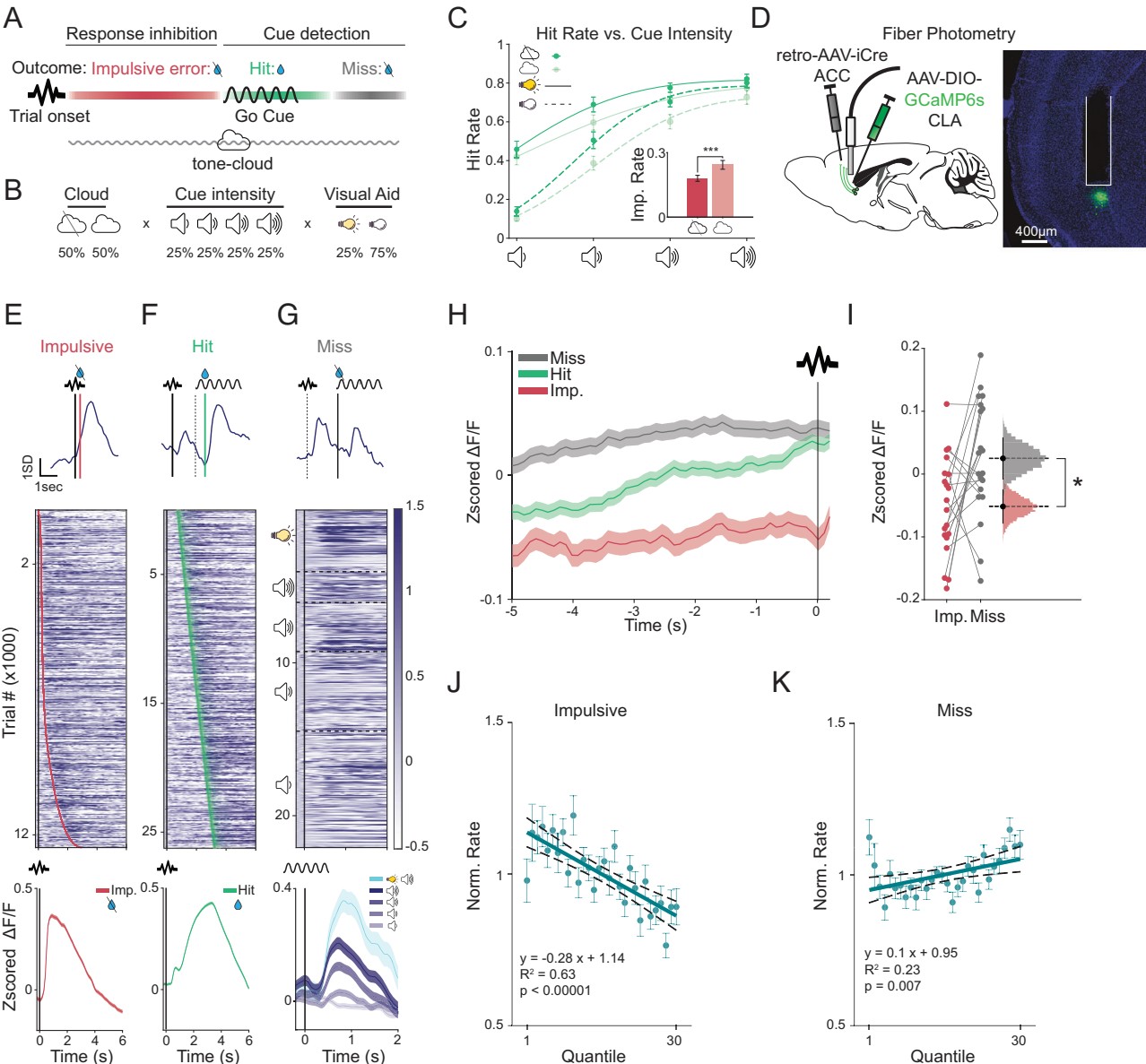

**Fig. 2 | ACCp recruitment in the ENGAGE task. A** The ENGAGE task requires mice to inhibit responses to distracting stimuli, and lick following a target cue. Trials are initiated with a brief broad-band noise tone ('No Go'), followed by an auditory 'Go' cue (pure tone pips). A random delay (0.5–3 s) separates 'No Go' from 'Go' cues. Following the Go cue, a response window (1.5 seconds) opens for a correct lick, rewarded with a drop of sugar water ('Hit'). Licks between trial onset and the Go cue ('impulsive errors') are not rewarded. Miss trials are also not rewarded. **B** Distribution of trial parameters. A tone-cloud (50% of trials) masks the 'Go' cue (presented at 4 different intensities); 25% of trials are assisted by a visual aid concurrent with the 'Go' cue. **C** Mean hit rate as a function of cue intensity. Inset: Effect of the tone-cloud on impulse errors ($n = 25$, $t_{(24)} = 5.583$, $p < 0.00001$, paired $t$-test). **D** Approach for fiber photometry recordings from ACCp with either GCaMP6s or jRGECO. Right panel depicts representative reporter expression and optic fiber placement. **E**–**G** $Z$-scored ACCp dynamics from 20 mice, during impulsive (**E**, $n = 12,471$), hit (**F**, $n = 26,243$) or miss (**G**, $n = 23,430$) trials aligned to trial onset (**E**, **F**) or cue presentation (**G**). Top: single trial examples. Solid line indicates alignment reference point (trial or cue onset). Heatmaps: ACCp activity across trials sorted by lick onset (impulsive); the delay from trial onset to cue (Hits); or cue intensity (Miss). The first impulsive (red) or correct (green) lick is marked for each trial. Bottom: mean activity traces in Impulsive (left), Hit (middle), and Miss trials (right, separated by cue intensity). **H** Pre-trial ACCp activity averaged over mice ($n = 20$), aligned to trial onset and separated by trial outcome. **I** Overall mean pre-trial activity preceding Impulsive (red) or Miss (gray) errors ($n = 20$, $t_{(19)} = 2.54$, $p = 0.0199$, paired $t$-test). Dots depict individual mice; histograms depict the bootstrapped distribution of means with 95% confidence intervals. **J** Normalized impulsive error rate, as a function of quantiles of ACCp pre-trial activity ($n = 20$, $p < 0.00001$; $R^2 = 0.63$, simple linear regression). **K** Normalized miss error rate, as a function of quantiles of ACCp pre-trial activity ($n = 20$, $p = 0.007$; $R^2 = 0.23$, simple linear regression). Thick lines represent linear fit, dotted lines represent 95% confidence intervals. Unless noted otherwise, data are presented as mean across mice ± s.e.m.

was higher than that of explorers, indicating that they were implementing a consistent behavioral strategy (Fig. S6A). Explorer mice, on the other hand, appeared to exhibit more fluctuations in their behavior, transitioning between states of high engagement (and impulsive errors) and low engagement (and misses). Consistent with transitions between engagement states, explorers tended to cluster miss trials into longer consecutive stretches than did exploiters (Fig. S6B). The reaction time in hit trials was similar in both groups of mice, suggesting that the difference in their performance could not be explained by differences in sensory-motor coupling (Fig. S6C).

Are these distinct behavioral strategies differentially reflected in ACCp activity and the relation between pretrial ACCp activity and

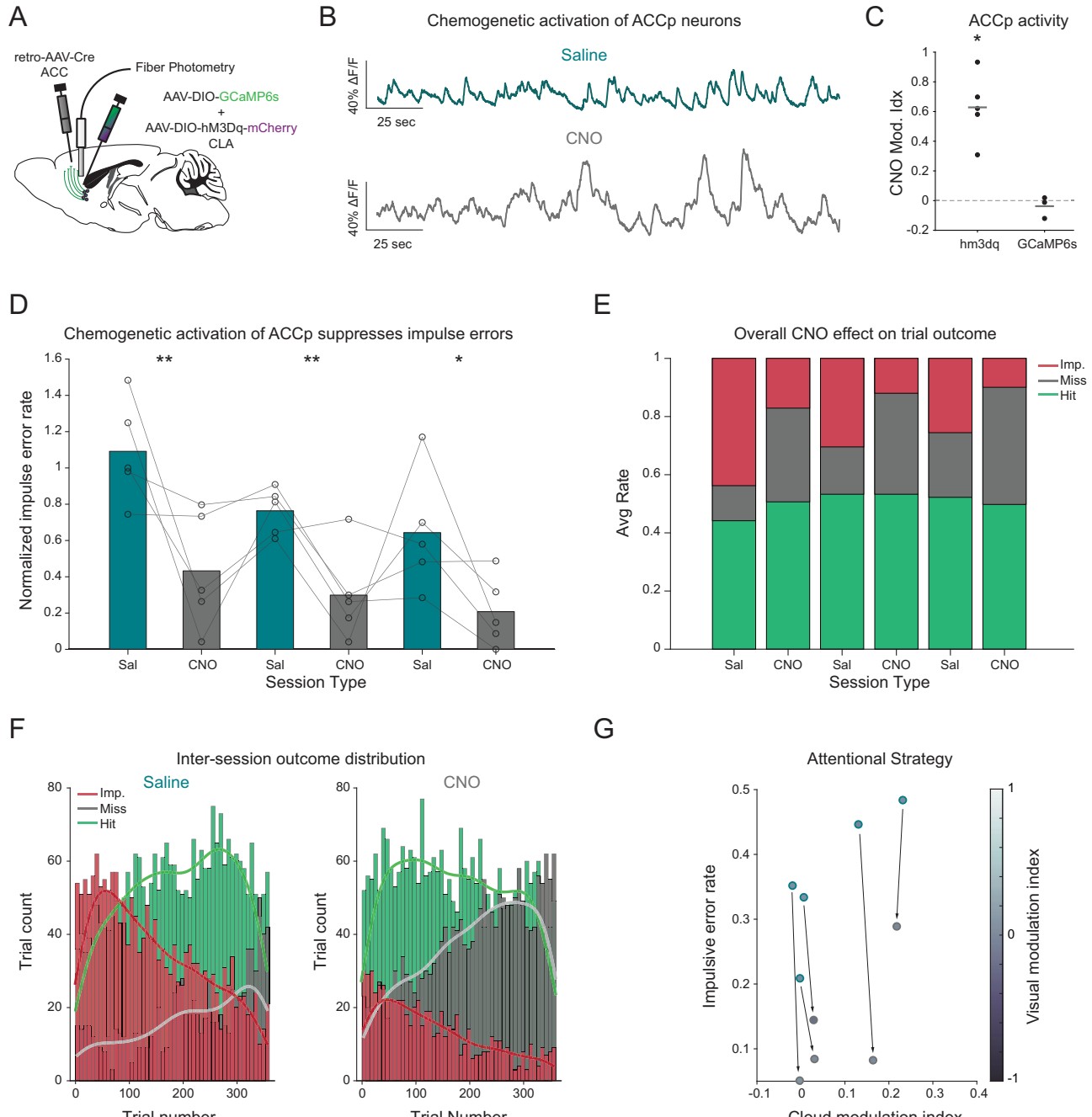

**Fig. 3 | Chemogenetic activation of ACCp activity promotes disengagement.**
**A** Experimental design for simultaneous chemogenetic activation and ACCp photometry. **B** Representative recordings of spontaneous ACCp activity in the same mouse following saline (top; teal) vs CNO (bottom; gray) administration. **C** Change in ACCp activity, quantified as a modulation index of change between saline and CNO (10 mg/kg i.p) sessions, in hM3Dq test mice ($n = 5$, $t_{(4)} = 6.244$, $p = 0.0034$, *student's t-test*) and GCaMP6s expressing controls ($\underline{n} = 3$, $t_{(4)} = -0.896$, $p = 0.465$, Student's *t*-test). Horizonal bar indicates mean, dots represent individual mice. **D** Impulsive errors in interleaved daily sessions of saline vs CNO ($\underline{n} = 5$, Saline, day 1 vs. CNO, day 1. $t_{(8)} = -3.436$, $p = 0.0044$, two-sample post hoc *t*-test. Saline day 2 vs. CNO day 2. $t_{(8)} = -3.642$, $p = 0.0033$, two-sample post hoc *t*-test. Saline day 3 vs. CNO day 3. $t_{(8)} = -2.533$, $p = 0.0175$; two-sample post hoc *t*-test. $F_{(1, 26)} = 9.33$, $p = 0.0052$;

$F_{(4, 26)} = 8.91$, $p = 0.0061$. Main effects of treatment and repetition, respectively, and no significant interaction, $F_{(4, 26)} = 1.11$, $p = 0.302$, *ANOVA on linear mixed effects model*). Bars indicate means, connected circles indicate individual mice. Error rates are normalized to the average rate over 3 prior days of saline habituation. **E** Average overall trial outcome across all mice over six experimental days. **F** Binned histograms (vertical bars) and kernel fit (smooth horizontal lines) of the distribution of trial outcome within saline (left) or CNO (right) sessions (3 sessions/each from $n = 5$ mice; 360 trials/session). **G** Impact of ACCp activation on individual mice behavior, plotted as a function of their impulsive error rates and the degree to which their performance was hindered by the presence of the tone cloud distractor (cloud modulation index) ($n = 5$; cyan outline: saline sessions, gray outline: CNO sessions). Circle shading represents improvement gained by the visual aid (visual modulation index).

performance? Indeed, explorers, which were susceptible to impulsive errors triggered by the 'No Go' BBN, exhibited a clear representation of this event in their ACCp signal. The representation of the BBN stimulus was absent from the ACCp signal of exploiter mice, who were less

prone to impulsive actions (Fig. 4E). Next, we investigated whether pre-trial ACCp activity showed differential associations with trial outcomes between these groups. Pre-trial activity in explorer mice from which we recorded ACCp activity ($n = 14$) showed a strong correlation with trial

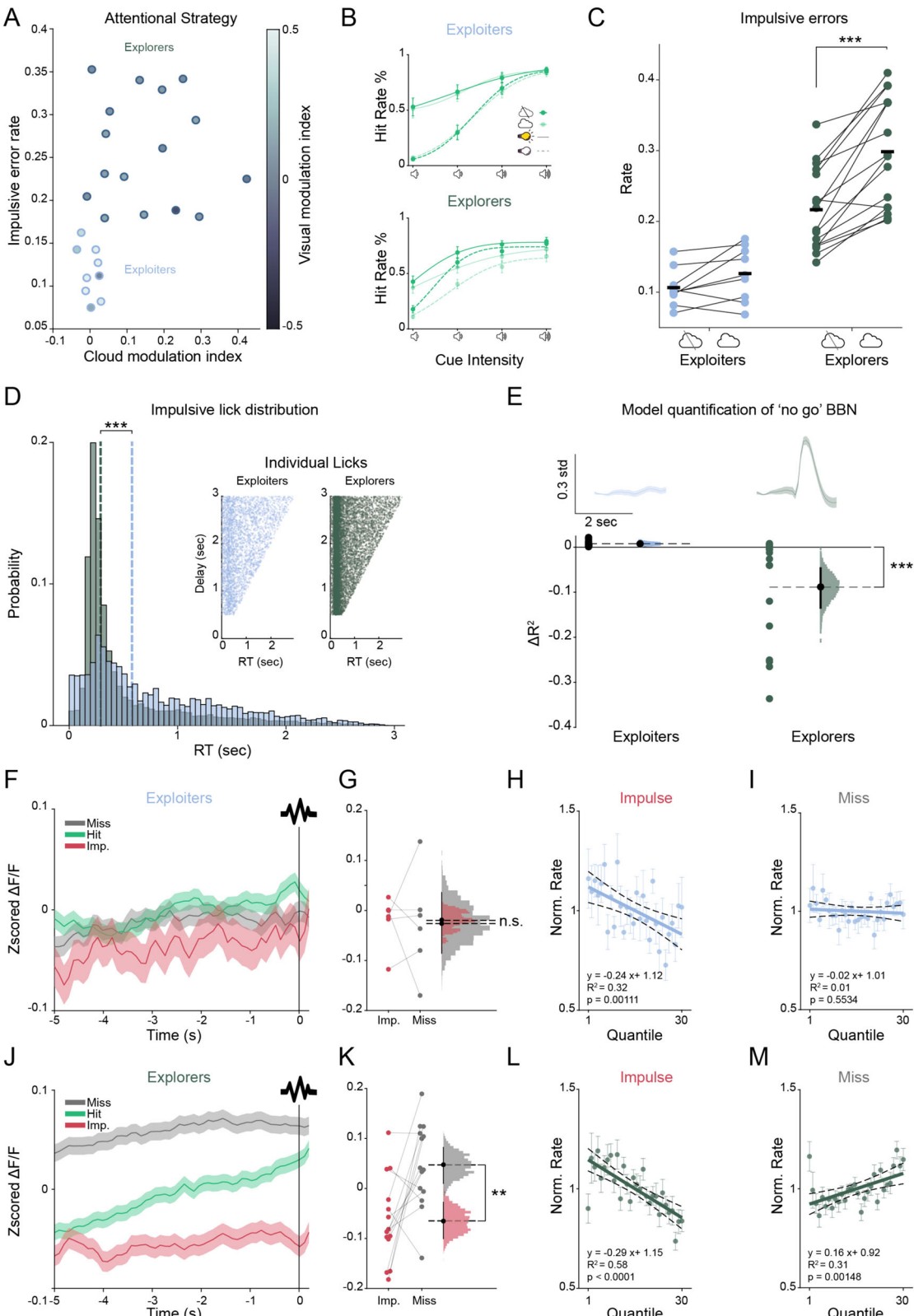

outcome (Fig. 4J, K). However, this separation was not evident in ACCp activity of exploiter mice ($n = 6$; Fig. 4F), in which pre-trial activity preceding misses and impulsive errors was on average indistinguishable (Fig. 4G). Pre-trial ACCp activity in exploiters still inversely correlated with impulsive errors (Fig. 4H), but this correlation was weaker than observed across the entire cohort, and no correlation was observed between pre-trial ACCp activity and misses in this group of

mice (Fig. 4I). These results affirm our hypothesis that the representation of task parameters in the ACCp signal is linked to the behavioral significance of those parameters. In mice that engaged with the full parameter space of the task we found that performance was closely correlated with ACCp pre-trial activity. Conversely, in a subset of mice employing an exploitative strategy, pre-trial ACCp activity was less correlated with performance, indicating that the establishment of

**Fig. 4 | ACCp recruitment corresponds to individual differences in behavioral strategy. A** Mice were split into two groups, explorers (dark outline, $n = 16$ explorers) vs. exploiters (light outline, $n = 9$), based on individual strategy plotted as a function of impulsive error rate vs cloud modulation index. Shading represents visual modulation index. **B** Mean hit rate as a function of cue intensity for each group. **C** Effect of tone-cloud on impulsive error rates by group ($n = 16$ explorers, cloud vs. no cloud $t_{(15)} = -6.37$, $p < 0.0001$, post hoc paired $t$-test. Interaction between cloud and group effects was significant, $F_{(4, 46)} = 11.97$, $p = 0.0012$, while individual main effects were not $F_{(1, 46)} = 1.89$, $p = 0.018$; $F_{(1, 46)} = 1.94$, $p = 0.017$, respectively. ANOVA on linear mixed effects model. **D** Distribution of impulse errors response times (RT) relative to trial onset. Dotted lines indicate medians. Inset: all impulsive licks relative to trial onset, sorted by the delay period (exploiters, $n = 3508$ trials; explorers, $n = 12,348$ trials). $D = 0.298$, $p < 0.00001$, Two-sample Kolmogorov-Smirnov test). **E** Evoked ACCp response to the trial onset BBN. Top: mean response. Bottom: unique contribution of trial onset to the ACCp signal (see "Methods" section) ($n = 6$, ACCp exploiters and $n = 14$ ACCp explorers.

$X^2(2) = 10.46$, $p = 0.0053$, Kruskal Wallis ANOVA on BBN UC by group, followed by post-hoc permutation tests on bootstrapped distributions $p < 0.0001$ ($0/5000 \leq 0$). **F** Mean pre-trial activity in ACCp recordings from exploiter mice ($n = 6$) aligned to trial onset and separated by outcome. **G** Mean pre-trial activity preceding impulsive (red) or miss (gray) errors in *exploiter* mice ($n = 6$, $t_{(5)} = -0.14$, $p = 0.891$, paired $t$-test). **H** Normalized impulse error rate, as a function of ACCp pre-trial activity quantiles in exploiter mice ($n = 6$, $p = 0.0011$; $R^2 = 0.32$, simple linear regression). (**I**) Normalized miss error rate in exploiter mice ($n = 6$, $p = 0.55$; $R^2 = 0.01$, simple linear regression). Thick line represents linear fit, dotted lines represent 95% confidence intervals. **J–M** Same comparisons as **H, I** for explorer mice (**K**: $n = 14$ mice, $t_{(13)} = 3.16$, $p = 0.0076$, paired $t$-test $n = 6$; **L**: $n = 14$ ACCp explorers, $p < 0.0001$, $R^2 = 0.58$, simple linear regression. **M**: $n = 14$ mice, $p = 0.0015$, $R^2 = 0.31$, simple linear regression). Unless noted otherwise, dots represent mean across mice, error bars represent s.e.m, histograms depict bootstrapped distribution of means with 95% confidence intervals.

a simplified behavioral strategy may uncouple performance from claustrum activity. These results demonstrate that mice which exhibit an explorative strategy, and are susceptible to impulsive errors in response to 'No Go' stimuli, exhibit a strong coupling of pre-trial ACCp activity with trial outcome. In contrast, mice displaying an exploitative strategy, focusing on only the most prominent cues, show weaker coupling between pre-trial ACCp activity and trial outcome.

## Discussion

In this study we found that the activity of claustrum neurons projecting to the anterior cingulate cortex serves as a defining factor for the engagement levels observed both in sleep and during task performance. Elevated ACCp activity promotes NREM sleep and reduces sound-evoked awakening from NREM sleep. Similarly, during performance in the ENGAGE task, elevated ACCp activity reduces impulsive errors and is correlated with higher miss rates.

Targeting claustrum neurons based on their projections is emerging as a key factor for their functional segregation[26,34,36,71,72], complementing differentiation based on genetic identity[29,30,32,73,74]. While a role for the claustrum in facilitating NREM sleep has been previously described[17], our findings implicate a particular claustral network comprising of ACCp neurons in this process, and demonstrate the functional effect of elevating the activity of ACCp neurons on reducing sensory-evoked awakenings. This logic was maintained in a demanding behavioral task, where ACCp activation decreased impulsive errors, and increased behavioral lapses. Importantly, many of our findings tying ACCp activity to engagement in both sleep and the ENGAGE task were not evident when recording from a population of claustrum neurons projecting to a different cortical target, the orbitofrontal cortex (OFC). In this context, it is worth mentioning that seemingly opposite effects on impulsivity have been observed upon manipulation of claustrum neurons projecting to different targets, such as the prefrontal cortex[31] and the somatosensory cortex[26]. We suspect, and our data supports, that distinct claustral circuits promote different functions. Thus, these and other incongruent effects previously described in the claustrum literature, are likely due to different, possibly opposing, functions of claustral projection networks. Understanding how genetic identity interacts with anatomy to define functional claustrum subnetworks should be prioritized in future studies.

The integration of fiber photometry and high-resolution behavioral analysis from a large cohort of mice afforded us a rare opportunity to observe how behavioral strategy corresponds to differential modulation of the activity of defined brain networks. One group of mice (exploiters) adopted a simpler, less cognitively demanding strategy to succeed in the ENGAGE task, by attending only the most prominent 'Go' cues (i.e. the loudest auditory cues and the visual aid). Doing so allowed these mice to essentially disregard the initial 'No Go'

BBN and reduced the negative impact of the tone cloud, which masked weaker auditory 'Go' cues. In contrast, most mice (explorers) remained susceptible to all sources of interference, while engaging with more trials overall, including the most difficult ones, at the cost of making more impulsive errors. We found that these approaches were distinctly reflected in ACCp activity. The representation of the 'No Go' BBN in the ACCp signal of explorer, but not exploiter mice is consistent with reports whereby the claustrum is selectively responsive to salient cues[22,28,75]. Furthermore, ACCp pre-trial activity was more strongly correlated with performance in explorer mice than in exploiters. It could be speculated that explorer mice had not yet settled on a simplified strategy of dealing with the task, and were still exploring its parameters, even after extensive training. Accumulating evidence supports a role for the claustrum during learning[25], but not for the expression of learned contingencies[29,30], and the claustrum has been found to be dispensable for the performance of a simple cue-response task on which mice were over-trained[74]. In the context of these published reports, it may further be speculated that upon commitment to a defined strategy, as may be the case in 'exploiter' mice, performance becomes more automated, and less associated with fluctuations of claustrum activity. In this study, we did not aim to overtrain mice, since we were interested in observing their claustral activity during errors. However, future work could address how claustral activity changes throughout learning and the acquisition of expertize, as behavior becomes more automated.

What might be the mechanism through which ACCp neurons restrict engagement? In mice, optogenetic stimulation of claustral projections to the ACC and frontal cortex elicits mixed excitation and inhibition, which includes prominent feed-forward inhibition, primarily of deep cortical neurons[17,30,33], but see also[50]. Such inhibition drives cortical slow wave activity (SWA) and may further contribute to sleep-like dynamics during wakefulness[17]. Indeed, previous studies have identified a role for the claustrum in regulating both REM sleep[16,76] and NREM sleep[17,18]. A recent study has further explored the state-dependence of the impact of the claustrum on cortex, observing that it is most prominent during NREM sleep, at which time optogenetic stimulation of the claustrum elicits SWA activity at the delta frequency and reduces gamma activity[77]. In this context, ACCp claustrum activity may promote deeper NREM sleep with less awakenings by synchronizing the activity of deep cortical neurons, giving rise to SWA.

Could this framework also explain the relationship between ACCp activity and performance in the ENGAGE task? The ACC has been prominently implicated in motor control and impulsivity, with recent work demonstrating that global network activity in the ACC gates response inhibition and is modulated as a function of task engagement[78]. How is cortical activity related to engagement? A bell-shaped curve between engagement and performance, known as the

"Yerkes-Dodson rule", associates moderate levels of engagement with optimal performance and both hypo- and hyper-engagement with detrimental performance[79]. Hypo-engagement leads to behavioral lapses and missed opportunities, whereas hyper-engagement leads to impulsive action in response to irrelevant cues[80,81]. A strong inverse correlation has been observed between engagement and 'idle' slow EEG rhythms such as SWA, first demonstrated in humans almost a century ago by electroencephalography (EEG)[82]. Indeed, hypo-engagement, defined by missed trials or delayed responses (lapses), correlates with increased SWA in humans and rodents[6,8,83]. In contrast, attention and arousal have been linked to cortical desynchronization, associated with a reduced SWA and increased high-frequency activities[9,10,84–86]. Together, our observations, synthesized with previous literature, suggest that ACCp neurons may restrict engagement by promoting SWA. Future work could investigate this possibility by combining cortical electrophysiology during task performance along with claustrum monitoring and manipulations[26].

Altogether, our results establish a connection between the activity of ACCp claustrum neurons and engagement, the fundamental ability to regulate how likely perception is to drive action. We find that ACCp neurons act to restrict engagement across diverse behavioral states, ranging from natural sleep to task performance. Fluctuations in ACCp activity contribute to the degree of behavioral lapses vs. impulsive errors and are associated with overall individual behavioral strategies favoring exploration. Our identification of a specific group of neurons linked to impulse control, along with their thorough transcriptional profiling[52], has the potential to open new avenues for therapeutic innovation in addressing various psychiatric and neurological disorders characterized by impaired impulse control.

## Methods

### Data acquisition

**Animals.** All mice described in this study were male C57BL/6JOLAHSD obtained from Harlan Laboratories, Jerusalem, Israel. Mice were housed in groups of littermates and kept in a SPF (specific pathogen-free) animal facility under standard environmental conditions: temperature (20–22 °C), humidity (55 ± 10%), and 12–12 h light/dark cycle, with ad libitum access to water and food. Mice were randomly assigned to experimental groups. All experimental procedures, handling, surgeries, and care of laboratory animals used in this study were approved by the Hebrew University Institutional Animal Care and Use Committee (NS-19-15584-3; NS-19-15788-3) and by the Institutional Animal Care and Use Committee (IACUC) of Tel Aviv University (approval 01-19-078, 01-20-028).

**Stereotactic surgery and viral injections.** Induction and maintenance of anesthesia during surgery were achieved using SomnoSuite Low-Flow Anesthesia System (Kent Scientific Corporation). Following induction of anesthesia, animals were quickly secured to the stereotaxic apparatus (David KOPF instruments). Anesthesia depth was validated by toe-pinching and manual heart-rate monitoring. Isoflurane levels were adjusted (0.8–1.5%) to maintain a heart rate of ~60 bpm. The skin was cleaned with Betadine (Dr. Fischer Medical); Lidocaine (Rafa Laboratories) was applied to minimize pain; and Viscotears gel (Bausch & Lomb) was applied to protect the eyes. An incision was made to expose the skull, which was immediately cleaned with hydrogen peroxide, and a small hole was drilled using a fine drill burr (model 78001RWD Life Science). Using a microsyringe (33GA; Hamilton syringe) connected to an UltraMicroPump (World Precision Instruments), a virus was subsequently injected at a flow rate of 50–100 nl/min, after which the microsyringe was left in the tissue for 5–10 min after the termination of the injection before being slowly retracted. For photometry experiments, a fiberoptic ferrule (400 μm, 0.37–0.48 NA, Doric Lenses) was slowly lowered into the brain. A custom-made metal head bar was glued to the skull, the incision was closed using Vetbond bioadhesive (3M), and the skull was covered in dental cement and let dry. An RFID (radio-frequency identification) chip (ID-20LA, ID Innovations) used for tracking during behavioral training was implanted subcutaneously. Mice were then disconnected from the anesthesia and were administered a subcutaneous saline injection for hydration and an IP injection of the analgesic Rimadyl (Norbrook) as they recovered under gentle heating. Coordinates for the claustrum were based on the Paxinos and Franklin mouse brain atlas[87]. Unless noted otherwise, viruses were prepared at the vector core facility of the Edmond and Lily Safra Center for Brain Sciences at the Hebrew University, as described previously[39]. See Supplementary Table 1 for injection sites and viruses used.

In mice undergoing sleep experiments, fiber photometry was complemented with EEG and EMG polysomnography. After viral injections at Hebrew U., mice were transported to Tel Aviv University for habituation and sleep recordings. To this end, during surgery two screws (one frontal and one parietal; 1 mm in diameter), were placed over the right hemisphere for the EEG recording, and two additional screws were placed above the cerebellum as reference and ground (in DREADD sleep experiments, frontal EEG was referenced to parietal EEG). EMG was measured via two single-stranded stainless-steel wires inserted to either side of the neck muscles in a bipolar reference configuration. EEG and EMG wires were soldered onto a custom-made headstage connector. Dental cement was used to cover all screws and EEG/EMG wires.

**Histology.** Mice were anesthetized for terminal perfusion by a ketamine/xylazine mix and perfused with RT PBS, followed by cold 4% PFA. Following decapitation, heads were placed in 4% PFA overnight to preserve the location of the optic ferrule. Brains then were carefully extracted and placed in 4% PFA for another night prior to transitioning to PBS in preparation for sectioning and staining. The fixed tissue was sectioned using a Vibratome (Campden 7000smz-2) at 60 μm thickness. To enhance GCaMP6s signals for analysis of axonal projections (Fig. S2), floating section immunohistochemistry was performed (rabbit anti-GFP, Life Technologies, Bethesda, MD; catalog No. A-6455; final dilution to 1:500 in 3% normal horse serum), following previously described protocols[39].

**Image acquisition.** Slides were scanned on a high-speed, fully-motorized, multi-channel epifluorescent light microscope (Olympus IX-81). Slices were imaged at 10X magnification (NA = 0.3). Green and red channels exposure times were selected for optimal clarity and were kept constant within each brain series. DAPI was acquired using excitation filters of 350 ± 50 nm and emission 455 ± 50 nm; eGFP, excitation 490 ± 20 nm and emission 525 ± 36 nm and Alexa 647, excitation 625 nm and emission 670 nm.

**Sleep monitoring with EEG and EMG.** Following surgery as described above, EEG and EMG were digitally sampled at 1,017 Hz (AM amplifier or PZ2 amplifier, Tucker-Davis Technologies) and filtered online: Both EEG and EMG signals were notch-filtered at 50/100 Hz to remove line noise and harmonics; then, the EEG and EMG signals were band-pass filtered at 0.5-200 Hz and 10-100 Hz, respectively. Due to a technical issue, EEG data in some animals were high-pass filtered in hardware > 2 Hz, but a comparison with full broadband (>0.5 Hz) EEG in other mice verified that this filter did not affect our ability to analyze sleep stages. Simultaneous video data (used for sleep scoring and for behavioral assessments) were captured by a USB webcam (at Tel Aviv University) and synchronized with electrophysiology / photometry data. Offline, EEG and EMG data were resampled to 1,000 Hz (MATLAB, The Math-Works) for sleep scoring.

**In vivo fiber photometry recordings.** Fiber photometry data was collected using a 1-site Fiber Photometry system (Doric Lenses,

Canada) adapted to two excitation LEDs at 465 nm (calcium-dependent GCaMP fluorescence) and either 405 nm (isosbestic control channel) or 560 nm (for two-color recording using jRGECO). Simultaneous monitoring of the two channels was made possible by connecting the LEDs to a 4-port minicube (with dichroic mirrors and cleanup filters to match the excitation and emission spectra; FMC4 or FMC5, Doric Lenses) via an attenuating patch cord (400 μm core, NA = 0.37-0.48). LEDs were controlled by drivers that sinusoidally modulated 560 nm/465 nm/405 nm excitation at 210/210/330 Hz, respectively, enabling lock-in demodulation of the signal (Doric Lenses, Canada). Zirconia sleeves were used to attach the fiber-optic patch cord to the animal's cannula. Data were collected using Femtowatt photoreceiver 2151 (Newport) and demodulated and processed using an RZ2 (at Tel Aviv University) or RZ5P (at the Hebrew University). BioAmp Processor unit and Synapse software (Tucker-Davis Technologies). LED intensities were modulated for each recording site to allow the recording of viable signals with the minimal intensity. To this end, each LED intensity was increased gradually until robust GCaMP/jRGECO fluctuations were observed above noise. Next, the 405 nm (isosbestic control channel) LED intensity was set to allow detection of motion artifacts. The total power at the tip of the patch cable was usually 0.05–0.1 mW with slight individual variability. The signal, originally sampled at 24,414 Hz, was demodulated online by the lock-in amplifier implemented in the processor, sampled at 1,017.25 Hz and low-pass filtered with a corner frequency at 4 Hz. All signals were collected using Synapse software (Tucker-Davis Technologies).

**Recording during undisturbed sleep.** Sleep experiments began six to eight weeks after surgery (due to the transfer of mice from the Hebrew University to Tel Aviv University and quarantine). Mice were placed in a new home cage within a double-wall, sound-attenuating acoustic chamber (Industrial Acoustics Company, Winchester, UK) and connected to the EEG/EMG headstage and to the fiberoptic patch cord. After >72 h of habituation to the new cage, to tethered recording and to the sounds described below in the tone-evoked arousal experiment, electrophysiology and photometry data were recorded continuously for 12 hours during light-phase daytime hours (starting around light onset 10AM) while the mice were undisturbed and behaving freely. To minimize bleaching and photo-toxicity, LEDs were automatically disengaged for 30 min every 2 h (90 min ON/30 min OFF, see Fig. S1D).

**Tone-evoked arousal experiments.** Experiments (lasting on average ~10 h, starting shortly after light onset 10AM) were conducted in the acoustic chamber. Sounds were generated using TDT software, amplified (SA1, Tucker Davis Technologies), and played free-field through a magnetic speaker (MF1, TDT) mounted 50 cm above the animal. In arousal experiments, BBN bursts (1 s duration, either 65 dB or 80 dB SPL as measured by placing a Velleman DVM805 Mini Sound Level Meter at the center of the cage floor, order randomized) were presented intermittently every 60 sec (±0.5 s jitter) when the mice were undisturbed. The sensitivity of the setup was confirmed by verifying that awakening probability was significantly higher for louder sounds (19.8 ± 8.4% for 80 dB SPL versus 8.1 ± 2.6% for 65 dB SPL. $n = 12$, $p < 0.001$, paired $t$-test). The analysis presented in Fig. 1E is based on the louder sound, for which there was a sufficient number of trials in both conditions (maintained sleep and awakening). Whenever COVID-19 lockdown restrictions allowed ($n = 9/12$ mice), we performed two separate experimental sessions per animal.

**Chemogenetic activation during sleep.** Two hours after light onset mice received an IP injection of either saline (vehicle control) or clozapine-n-oxide (CNO), diluted to a final dilution of 1 mg/ml (10 mg CNO in 500 μl DMSO and 9.5 ml saline) and administered at a dose of 10 mg/kg. Mice were then recorded for 4 hours either undisturbed or

under the tone-evoked arousal protocol. Analysis was later carried on the 2-hour recording segment starting 30 mins after injection.

**ENGAGE behavioral task.** Training in preparation for fiber photometry recording in head-fixed mice was performed in automated behavioral cages (see *Automated behavioral training* below). Subsequently, trained mice were habituated to head-fixation and performance of the ENGAGE task while head-fixed but free to run on a linear treadmill (Janelia 2017-049 Low-Friction Rodent-Driven Belt Treadmill). As described in Fig. 2A, B, each trial in the task is initiated with a brief auditory broad-band noise stimulus to which the mice must refrain from responding ('No Go'), followed by an auditory 'Go' cue (a set of five 6 kHz pure tone pips). A delay period of random duration (0.5-3 seconds) separates the 'No Go' from the 'go' cues. Licks during this delay period (i.e. between the 'No Go' and the 'Go' cues) result in the termination of the trial and label the trial result as 'impulse error'. Following the 'Go' cue, a 1.5-second-long response window is open for a correct lick ('hit'). Absence of licking within the defined response window labels the trial as a 'miss' and is not rewarded. 'Hit' trials are rewarded with a single drop (4 μl) of sucrose water. Trial difficulty was determined by a combination of several factors: four equally probable intensities of the 'Go' cue tone, inclusion of a tone-cloud 'mask' (4 seconds of continuous chords assembled from logarithmically spaced pure tones in the frequency range of 1-10 kHz, excluding the Go cue frequency, in 50% of trials;), and, in 25% of trials, a visual stimulus ('visual aid') presented concurrently with the 'Go' cue. Trials were initiated automatically every 20 seconds. Behavioral sessions contained blocks of trials containing 15 occurrences (in some cases shortened to 8 or 10) of each possible combination of parameters, in random order. These blocks were repeated 2–4 times (as long as mice maintained participation) for a total of up to 1,000 trials per mouse per day. (A typical daily session totaled 240-480 trials). The degree to which the different trial parameters (cloud, visual aid) affected behavior (impaired or improved it, respectively) was quantified by calculating a modulation index for the effect of each parameter on hit rates in the task (i.e., difference normalized by sum: $Idx(A,B) = \frac{hit\_rateA - hit\_rateB}{hit\_rateA + hit\_rateB}$).

**Automated behavioral training.** To train mice on the ENGAGE task, we developed a custom automated behavioral setup: Connected to the home-cage, this setup supports self-initiated simultaneous individualized training of co-housed mice in their home environment. Automation was implemented solely for the purpose of accelerating training, removing the dependency on the recording rig, and increasing the experimental throughput. Any direct comparison between the training and head-fixed conditions is beyond the scope of the current work. Training cages comprised a 4 cm diameter tube corridor connected to the home cage of the mice. A behavioral lick port (Sanworks) was positioned at the end of the corridor. Within the training cage mice had ad libitum access to food, while access to water was restricted to the output of the behavioral system. A radio-frequency identification (RFID) reader (ID-20LA, ID Innovations) was positioned above the corridor for individualized identification of mice. Auditory cues were delivered by a Bpod wave player (Sanworks) connected to earphones positioned on the corridor adjacent to the port. Experiments were controlled via an open-source, MATLAB-based state machine (Bpod, Sanworks). A custom protocol was written in MATLAB in order to support individualized training by gating the Bpod state machine as a function of the output of the RFID reader. This enabled activation of different task parameters for individual mice based on their performance. Mice were initially taught to associate an auditory-visual cue with water availability during a lick adaptation period. Further training comprised several stages of increasing difficulty towards the full task (see ENGAGE behavioral task above) and each mouse progressed individually, according to its learning. Entry of a mouse into the port

(an RFID reading) initiated a trial, as described above (see *ENGAGE behavioral task*). Impulsive or late licks were not rewarded, and mice had to exit the port (terminate and reinitiate RFID reading) before a new trial could be initiated. After mice reached satisfactory success rates (50–70% correct; 2.6 days on average; Stage 1) they proceeded to Stage 2, in which the random delay period was prolonged to 0.5–2 s (2.5 days on average). Mice then proceeded to Stage 3, which included the full range of possible delays (0.5–3 s) and a gradual transition to auditory trials with no visual aid (Aud), This transition occurred over three steps: 30% Aud (Stage 3a), 50% Aud (Stage 3b), and 70% Aud (Stage 3c). Following Stage 3 (4.4 days on average) a pure tone-cloud masking stimulus was introduced (4 seconds of continuous chords assembled from logarithmically spaced pure tones in the frequency range of 1-10 kHz, excluding the Go cue frequency; intensity = 67.5db SPL; Stage 4). Finally, we gradually added three attenuations of the target cue (Stage 5). Automated training took on average 29.6 days. Response duration and reward size were kept constant throughout training.

**Chemogenetic activation in the ENGAGE task.** To avoid introducing stress to the mice in the context of the *ENGAGE* task, injections were done 30 min prior to head-restraining them. Mice received an IP injection of either saline (vehicle control) or clozapine-n-oxide (CNO), diluted to a final dilution of 1 mg/ml (10 mg CNO in 500 µl DMSO and 9.5 ml saline) and administered at a dose of 10 mg/kg. Analysis of the ACCp signal and performance in the ENGAGE task encompassed the full session (360 trials) following IP injection.

## Data analysis

**Sleep staging and awakenings.** Sleep scoring was based on a combined manual examination of frontal and parietal EEG, EMG, and video data, using custom MATLAB user interface adapted from Sela et al.[88], which enabled precise marking of state transitions. Data were first divided to three vigilance states: (1) wake: low-voltage high-frequency EEG, accompanied by phasic EMG activity and behavioral activity (e.g., eating, grooming or exploring) confirmed with video; (2) NREM sleep: high-amplitude slow-wave activity and low tonic EMG activity; (3) REM sleep: high-frequency wake-like frontal EEG co-occurring with theta activity in parietal EEG and flat EMG. Examples of hypnogram, state pie chart and electrophysiological traces are shown in Supplemental Fig. 1A, B. Joint distributions of EMG levels and EEG high-/low-frequency power ratio in supp Fig. 1A were calculated based on 4 second epochs. Power ratio was defined as power >25 Hz divided by power <5 Hz in the frontal EEG. For sensory-evoked awakening data, each experimental trial was visually inspected and those occurring during sleep were categorized as either eliciting behavioral awakening or maintained sleep. Behavioral awakening was declared if wake-like EEG flattening (without dominant theta) was present within 3 s from sound onset and lasted for at least 2 s, and this EEG desynchronization was accompanied by EMG activation. All other trials were categorized as maintained sleep. To avoid adding trials in which sleep was not purely maintained (e.g., brief awakening, or EEG only awakening) trials with lowered SWA (<10 Hz) following sound were automatically taken out of analysis if the ratio between SWA 5 s before onset and 5 s after onset was greater than 1.5. Baseline analysis (of average ACCp ΔF/F) of auditory trials was performed on the 4 s baseline before sound onset.

**Fiber photometry analysis.** Unless otherwise noted, all analysis was performed using custom MATLAB scripts. First, to correct for baseline drift due to slow photobleaching artifacts (particularly during the first several minutes of each session), a 5th-order polynomial was fit to the raw data and then subtracted from it. After baseline correction, ΔF/F was computed using the 1st lowest percentile value as F0 ($\frac{\Delta F}{F} = \frac{F-F_0}{F_0}$), and the resulting trace was *z*-scored relative to the mean and standard deviation of the entire recording session to normalize across channels,

indicators, and mice. Due to the head-restrained configuration, motion artifacts were rare. However, for 2/30 mice, motion artifacts were corrected by using the *z*-scored isosbestic control channel as a sample-by-sample F0 for computing ΔF.

Within the ENGAGE task, to correct for small session-to-session fluctuations in the signal while maintaining quantitation of pre-trial activity, we calculated pre-trial activity for every trial (4 sec before trial onset) and used the pre-trial signal as a dependent variable in a linear model with recording session and trial outcome as independent variables (baseline ~ outcome + session). A scalar value of the intercept and estimate for each session was then subtracted from the corresponding data set, setting the mean baseline for correct trials in each session at approximately zero. Pre-processed data then was cut into 20-second windows (-5:15 sec) around each behavioral epoch (trial onset, cue onset, lick onset, and run onset) and concatenated for each mouse to form an event-aligned activity matrix, together with an information table detailing the parameters and outcome of each trial. Importantly, trials that contained impulsive licks within the first second of the trial were excluded from the analysis of sensory responses, to rule out a potentially confounding lick response.

For sleep recordings, to account for LED onset and offset every 90 minutes, the signal was segmented, and each part had separate preprocessing as mentioned above.

**Average state activation.** Average state activation as shown in Fig. 1C was calculated on undisturbed ~10-hour recordings following manual scoring. For each mouse the ACCp activation was averaged across state epochs, and the average between mice was calculated.

**Tone-evoked arousal baseline analysis.** Baseline was calculated as the 4 s average prior to sound onset and averaged within state in each recording block for statistical analysis. Given that different animals contributed different number of trials in this experiment, we weighted the relative contribution of each animal's data in the average according to its number of trials. The weights per mouse were calculated as #trials / average # of trials across mice.

**Linear encoding model.** A linear encoding model was constructed, using ridge-regression to create time-averaged kernels for each behavioral epoch in the task, following code from[64]. For each mouse all trials from all sessions were used to create the full model. Ten-fold cross-validated estimation of the explained variance by the full model (CVR$^2$) was then compared to that explained by each individual label by itself (single variable model). The unique contribution of each variable was estimated by the loss in explained variance ($\Delta R^2$) following omission of each variable from the full model. Both measures were normalized to the size of the window (Supplemental Table 2) and to the CVR$^2$ of the full model. Bootstrapping (www.estimationstats.com) was performed based on the work described in ref. 89.

**Histological quantification and statistical analysis.** In order to quantify labeled cells (number and overlap), automated image analysis was used. For each claustrum, we captured seven images from 60 µm thick slices, separated by 360 µm, which spanned Bregma −1.06 mm to Bregma +1.1 mm. The claustrum was manually cropped according to the outline depicted in the appropriate section from the Paxinos and Franklin mouse brain atlas[87]. The cropped image files were used in the analysis pipeline, with three channels for each image: DAPI, eGFP, and tdTomato. The data was analyzed using the CellProfiler v.3.0.0 co-localization pipeline (www.cellprofiler.org) with minor modifications, including feature enhancement and shrink/expand objects[90,91]. Fluorescently labelled retroAAV-H2B (6 ACC/OFC. See also Supplementary Table 1) expressed in the nuclei, allowing the detection of labeled nuclei directly from eGFP and tdTomato channels. Object-centroid distances were measured and calibrated so only objects with a

maximal centroid distance of 6 pixels were considered to be double-labeled cells. Histograms corresponding to the spatial localization of the labeled cells were built in RStudio (Ver. 1.0.153).

## Reporting summary

Further information on research design is available in the Nature Portfolio Reporting Summary linked to this article.

## Data availability

We have uploaded the data acquired during the development of this project to the following Zenodo repository: https://zenodo.org/uploads/10975842.

## Code availability

Code for reproducing the figures in this manuscript, when applied to the data deposited to the Zenodo repository mentioned above, is available on GitHub (github.com/Citrilab/Atlan_et_al_2024).

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

## Acknowledgements

The authors thank members of the Citri and Nir lab for their intellectual contributions throughout the multiple metamorphoses of this work. We also thank Profs. Inbal Goshen and Mickey London for comments on earlier versions of the work. This work was carried out with the support of the following funding sources: European Research Council (ERC 770951) (AC). European Research Council (ERC 864353) (YN). Israel Science Foundation (1326/15 & 51/11) (YN). The Israel Science Foundation (1062/18, 393/12, 1796/12, and 2341/15) (AC). The Israel Anti-Drug Administration (AC). EU Marie Curie (PCIG13-GA-2013-618201) (AC). National Institute for Psychobiology in Israel, Hebrew University of Jerusalem Israel founded by the Charles E. Smith family (109-15-16) (AC). Adelis Award for Advances in Neuroscience (AC). The Brain and Behavior Foundation (NARSAD 18795) (AC). German–Israel Foundation (2299-2291.1/2011) (AC). Binational Israel–United States Foundation (2011266) (AC). Milton Rosenbaum Endowment Fund for Research in Psychiatry (AC). Prusiner-Abramsky Research Award in Basic Neuroscience (AC). Seed grant from the Eric Roland Fund for interdisciplinary research (AC). Canadian Institute for Advanced Research (CIFAR) (AC). Contributions from anonymous philanthropists in Los Angeles and Mexico City (AC). Figure 1A, F were created with BioRender.com released under a Creative Commons Attribution-NonCommercial-NoDerivs 4.0 International license.

## Author contributions

Conceptualization: G.A., Y.N., and A.C. Methodology: G.A., N.M., N.P.R., I.Y., N.Z., E.C., T.K., N.B., E.S., M.G., Y.N., and A.C. Investigation: G.A., N.M., N.P.R., I.Y., N.Z., E.C., T.K., N.B., and E.S. Visualization: G.A., N.M., I.Y., N.Z., E.C., T.K., N.B., and E.S. Funding acquisition: Y.N. and A.C. Project administration: A.C. Supervision: Y.N. and A.C. Writing – original draft: G.A. and A.C. Writing – review and editing: G.A., N.M., N.Z., Y.N., and A.C.

## Competing interests

The authors declare no competing interests.
