## [Peer Review File · Nature Communications]

Clastrum neurons projecting to the anterior cingulate restrict engagement during sleep and behaviorReviewers' comments:

Reviewer #1 (Remarks to the Author):

The authors perform calcium fiber photometry in the claustrum in a novel sensory response task. Interestingly, they find that the claustrum is most active during lapses in engagement, during trial misses. This is a very important result. The paper is timely as another recent study was published last year showing claustrum neuron modulation on a multimodal sensory detection task. This new paper provides complimentary data and extends the new idea that claustrum may be involved in disengagement (an idea that is controversial, but nonetheless supported by the data). I have some comments on the analysis, and writing, but otherwise the paper is very interesting.

In Figure 1, is it the case that slow wave activity is also decreased prior to the awakening events? The impression is given that the reduced claustrum activity is 'causal' in the awakening, but perhaps the state of the mouse is generally different and claustrum activity simply covaries with activity in the cortex? Perhaps there is a way to determine if the variability in claustrum activity can account for the differences in awakening versus not (above and beyond that information provided by slow wave activity). Or maybe these 2 variables are too intimately linked, or too few trials to be able to conduct the analysis. In any case some further analysis or discussion of the interpretation is warranted.

I do not really understand the significance of the main effect of mouse identity in Figure 1E. The last sentence in paragraph 1 of the results mentions a positive linear correlation with respect to the quartiles of claustrum activity, but I do not see this reported in the figure legend or text. What are the units for Figure 3C on the y axis? In Figure 3, I find it odd that the signal fluctuations are compared between the 2 treatments rather than measuring how much each treatment impacted the signal independently, relative to the time period before each injection (as signals can vary across days even in a single mouse). For example, why not compare baseline versus CNO in 1 session, then baseline versus saline in another session (then compare the change from baseline between saline and CNO).

In Figure 3 the relative error rate is shown for CNO and saline. The critical control experiment is shown in Figure S7, but in this case the y-axis is labelled impulsive error rate. Why not show the same measurement between the experiment and control? Also, in Figure S7, perhaps the trial type can be broken down into hits, misses, and impulsive. The dose of CNO is very high, so these controls are important here. I'm not sure the 3 control mice are particularly compelling given that it would be hard to show any significant difference with and n=3.

In Figure 3 the authors can provide further support of their hypothesis, because they are measuring ACCp activity during the behavior. I would recommend determining how ACCp activity is organized on hit, miss, and impulsive trials with hm3d-CNO, as this would provide further evidence of the role of these neurons in task performance.

Related to the above point, when discussing the CNO results, the authors say that CNO did not impact task associated responses in the ACCp signal, but I cannot see any statistics to support this (Figure S7E). Also, with such a low sample size, I suspect detecting significance between these conditions would be challenging. Consider restating this in the results if statistical comparisons cannot really be drawn here.

Is there any indication if the 5 mice receiving hm3d are exploiters or explorers according to the criteria in Figure 4A?

In lines 283-285 the authors state that a simplified response strategy may lead to independence from the claustrum implying that this difficult task is somehow recruiting the claustrum. When in fact their data are leading me to believe that claustrum activity does not help the mouse in this task very much at all. Increased claustrum activity in explorers seems to correlate with misses, whereas effective performance (assessed by hits) may or may not correlate with moderate levels of claustrum activity (the authors do not correlate claustrum activity with hits). The word "recruited" implies that claustrum activity is playing a role in performing the task, but again, I do

not see the evidence for this. Please help me as a reader overcome this confusion, or please clarify the language. Perhaps my confusion is rooted in the fact that they state that exploiters use a simplified response strategy (and do not have different claustrum levels pre-trial), but how can the authors make this claim regarding what strategy is simple versus hard? It seems both strategies have their merit.

In the discussion, the authors state "our results demonstrate a causal role for pre-trial claustrum activity...". This is imprecise. The causal experiment does not analyze pre-trial activity. How do we (as readers) know that the hm3d manipulation can be explained by the effects on pre-trial activity at all, or specifically pre-trial activity. Sentences like this, start to imply that perhaps loss of function experiments or temporally specific (optogenetic) experiments should be done. I suspect the authors do not want to add these experiments to this paper. So, I would recommend limiting sentences that overstate the reach of the results.

Another part of the discussion mentions the decrease in impulsive behavior with the hm3d (Line 313). However, the authors do not mention the prominent increase in misses with hm3d-cno. These hm3d mice are not doing particularly better on the task (if anything, their hit rate goes down). Perhaps this should be highlighted as well in the discussion, as this is also a major finding in the paper (the correlation between increased claustrum activity and misses).

Reviewer #2 (Remarks to the Author):

In this study, Atlan and colleagues tested the hypothesis that the claustrum may regulate engagement. They recorded different populations of claustrum neurons either projecting to the anterior cingulate cortex (ACCp) or to the orbitofrontal cortex (OFCp). They suggest that ACCp neurons, but not OFCp ones, are more active during sensory disengagement during sleep. In addition, using a novel behavioral assay, the authors showed ACCp activity was high when mice were hypo-engaged and prone to behavioral lapses and was low when mice were doing impulsive responses. Chemogenetic elevation of ACCp activity suppressed impulsive behavior and reduced task engagement.

I have a mixed feeling about the study. On one hand, the hypothesis tested and some of the data are interesting. On the other hand the paper is difficult to read (mostly because it's not well structured, important information being scattered all over the place) and suffers from missing experiments/controls, possible alternative interpretation of the data and technical limitations. In its present form, the paper is in my opinion not acceptable. The authors will need to revise the study in depth, answer to the different comments and rewrite the manuscript (expand the text, reorganize and increase the number of main figures) before being published. However, if the authors address the criticisms, I have no doubt the paper will be important for the neuroscience community at large and would deserve publication in Nature Communications.

See specific comments below:

1- Characterization of different groups of claustrum projection neurons

The entire study relies on the analysis of claustrum neurons, labelled with retroAAV-cre, specifically projecting to either ACC or OFC areas. I have no doubt that the authors are claustrum experts, but I am quite confused by the way claustrum is defined in this study and by the quantification done. Although the authors showed some characterization of the labelled populations in Fig. S1, it is clearly not enough to ascribe cells to the claustrum and not the insular cortex. Undoubtedly, a much better characterization of the labeled populations is needed to interpret the results and support the claims of the study. See specific comments below:

- From what I can see in the Allen brain atlas, the claustrum is described as a small oval island (of ~300-600 microns in diameter) in the agranular insular cortex. Therefore, claustrum size and shape used by the authors seem to be quite different from the one defined by the Allen Institute.

As an example, H2B-tdTomato labelled neurons are distributed over more than 2mm in the DV axis in Fig. S1B, which seems too much to be only the claustrum. How can the authors exclude the targeting of insular neurons? For example, some insular neurons display partially comparable projection patterns (see https://connectivity.brain-map.org/projection/experiment/siv/296048512?imageId=296048530&imageType=TWO_PHOTON, SEGMENTATION&initImage=TWO_PHOTON&x=17806&y=11081&z=3; or https://connectivity.brain-map.org/projection/experiment/siv/485847695?imageId=485847813&imageType=TWO_PHOTON, SEGMENTATION&initImage=TWO_PHOTON&x=16509&y=14985&z=3; https://connectivity.brain-map.org/projection/experiment/siv/166153483?imageId=166153633&imageType=TWO_PHOTON, SEGMENTATION&initImage=TWO_PHOTON&x=24672&y=15052&z=3).

The authors could probably use antibodies against known claustrum and cortical markers in order to characterize the different populations targeted by the two retrograde injections.

I also noticed that the authors used in a previous publication a transgenic line with specific cre expression in claustrum neurons. How comparable are the cells labelled with the retroAAV and the ones found in the mouse line?

- In line with the previous comment, can the authors provide a clear definition of what they consider claustrum core and shell. From what I could read in the literature, it seems that PV staining is usually used in this field to distinguish the two. However, on the image shown on Fig S1D, I noticed ACCp neurons are found mainly outside the core, though it might be a problem of image quality in the pdf file. Similarly, the magenta signal seems to surround a core at bregma 1.1 and -0.7 in Fig. S1F.

- There seems to be much more OFCp than ACCp cells (Fig. S1B,C). However, the density plots give a different view. Unless I missed the information, I couldn't find any quantification regarding the number of labeled neurons in the two experimental conditions. The authors should plot this information at the different bregma levels since this might impact the interpretation of the imaging/behavior results (see below). The authors should also explain how the densities shown in Fig. S1C,E were computed.

- The authors used the spatial distribution of the two populations to make the claim that they are different. It might be true but the data are not compelling especially since the fraction of double labeled neurons remains relatively small when injections are made at the same coordinates (=same cortical area, for example 15% for ACC using nuclear-targeted fluorescent proteins which is less ambiguous, top panel Fig S1G,). To convince the reader, in addition to the characterization mentioned above, the authors should plot the same distributions displayed in Fig S1C,E for the experiments where double ACC or double OFC injections were made. How do these distributions compare to the ones already reported in Fig S1?

- In line with the previous comment, the authors used the axonal spread of the two populations to make the claim that they are different (Fig S1H,I). Are the axonal distribution really different (please plot individual data points and provide statistical tests)? I agree that ACCp mice show more axons in ACC than OFCp mice (and vice-versa). However, were the same quantification done for experiments with double injections in the same cortex? How variable are these results?

In addition, I did not understand the way the axonal quantification was done. In Line 478-483, you wrote "a manual threshold was set for every brain such that the claustrum area would be saturated and background would be minimal, enabling a clear contrast for the fluorescent processes. Structures of interest were selected based on previous anterograde tracing studies in the claustrum (27). Analysis was conducted in Fiji (ImageJ) and quantified as the mean pixel intensity in a rectangle sampled within different brain divisions. Measurements were obtained from qualitatively similar positions in each section over mice". How was the threshold decided and how much is this changing the results if varied? How can different brain slices be analyzed? What is the background level? Finally, are we looking at average pixel intensity in each area analyzed or have the authors used any form of normalization?

Minor comments:

- Fig S1C left panel: Could the authors plot a graph with actual distances instead of a.u.? What is the reason for using a.u.?
- Can the authors indicate which PV antibody was used for histology. Was the alexa647 (Line 452) used for this staining?
- Fig S1E the colors used to label the different cells are too similar to be discriminated even on a computer screen. I suggest using other colors (why not green and magenta?)
- Can the authors specify which experiment shown in Fig S1 use a PV-cre mouse (as mentioned in Suppl. Table 1).
- Fig S1I: AOB should be changed for AON
- What kind of image acquisition system was used? Confocal or epifluorescence microscope?
- Fig. S1I: Have the authors excluded prelimbic cortex from the analysis or combined it with rACC? The authors should also define in the methods how rACC, mACC, cACC, CVO, rVO were defined.

2- Calcium imaging using fiber photometry

The authors used photometry to specifically monitor activity from ACCp or OFCp neurons. Since photometry records an averaged population signal, it is likely impacted by the number/distribution of labeled neurons, and might not even necessarily report firing activity (see below). Since the authors claim that ACCp and OFCp neurons behave differently, important questions still need to be addressed before trusting the calcium data. See specific comments below:

- The authors used calcium photometry signals as a proxy for firing activity. However, a recent report convincingly showed that photometry don't primarily reflects somatic changes in calcium, at least in the striatum (see the publication DOI: 10.1038/s41593-022-01152-z). Since the entire study correlates claustrum "activity" to behavior, it is important to understand the source of the recorded signals (spiking, neuropil, dendritic information...). Can the authors estimate the number of imaged soma? Can the authors exclude that differences between ACCp and OFCp response represent differences in nonsomatic calcium responses (for example, ACCp may differ in their dendritic and/or axonal arbors)? Could the authors backup some of their claims with miniscope imaging or extracellular recordings?
- In line with previous comments, how does the number and the distribution of labeled neurons, as well as the optic fiber position, influence the photometry conclusions? For example, considering that more OFCp neurons seem to be labelled with retroAAVs and since photometry represent an averaged population signal, it is possible that specific neural responses from a subpopulation (especially if insular cortex and claustrum neurons would be simultaneously targeted, see section 1) are drown in OFCp signals but less in ACCp signals. As a second example, it seems that the authors imaged activity from caudal claustrum where there are less cells (RostroCaudal 0 from Suppl. Table 1 and as seen by the few labeled neurons in Fig. S7A). Would the results be different for more rostral location where there are more neurons? It is important to have such information since chemogenetic experiments targeted also frontal claustrum (RostroCaudal +1). Finally, as seen in Fig. S7A, the number of labelled neurons can be quite low and the position of the optic fiber lateral to the recorded field, is this kind of recording giving calcium signals?
- I found all the experimental configurations used in this study very confusing. There are multiple calcium indicators used (GCaMP6s and RGECO1) and neuronal targeting was done using either flp or cre-dependent viruses. Photometry recorded either a single signal or multiple signals at once (including ACC, OFC and even auditory cortex, and input/output dual imaging). I acknowledge that the authors honestly reported all the different experimental variations in Supplementary table 1. However, most of these conditions are neither discussed nor used in the manuscript. Furthermore

it's nearly impossible evaluating the possible confounding factors introduced by such diversity and the low experiment numbers for some configurations. Can the authors demonstrate the reproducibility of their findings irrespective of the configuration/indicators used.

Can the authors also show examples and quantification for the experiments in which ACCp and OFCp were simultaneously recorded. Are differences between AACp and OFCp activity still observed in such cases?

Was the dual GCaMP6s and jRGECO1 imaging truly simultaneous? How was the motion artefact corrected in this case?

- Can the authors exclude that photobleaching and/or phototoxicity play a role in the interpretation of the results for long recordings? For example, is the relationship between claustrum activity and either sleep or behavior during the ENGAGE task the same between the beginning and the end of the sessions.

Minor comments:

- Fig2G legend mentioned n =20 mice but I count 19 from the suppl table. Please clarify.

- Lines 556-557: you wrote "To this end, the 465nm LED intensity was increased gradually until robust GCaMP/jRGECO fluctuations were observed above noise."

Are you using the same wavelength for RGeco as well or is it a typo? Was the power chosen arbitrarily or was a specific criterion used?

3- Claustrum activity and sleep

I found the relationship between calcium activity and sleep/wake state intriguing but also confusing and correlative. On top of the questions raised in the previous two sections, additional analyses and experiments are needed for this section in order to validate some of the conclusions. See specific comments below:

- The link between depth of sleep, cortical responsiveness and claustrum activity is a bit confusing to me. Are we not expecting that claustrum activity should be stronger during REM than during non-REM sleep since REM sleep should be a light sleep state where it should be easier to wake up the mouse? Have the authors observed any relationship between sleep maintenance and photometry during REM?

- The correlation between the depth of NREM sleep and fiber photometry is not really convincing. Though I can acknowledge that a correlation might exist, additional analyses and/or experiments must be performed to secure this claim. For example, the relationship plotted in Fig. 1G is extremely variable and some mice are plotted multiple times. The authors should show which data points are coming from the same animal. If they would average the data between animals (n=6) and not between experiments, would the relationship still be significant? Have the authors observed any effect of photobleaching? Is the difference seen in Fig 1G observed at different periods of the experiments (early v. late recording periods)?

- This part of the study is unfortunately only correlative. In order to be causal, reduce the confusion mentioned above and strengthen their claims, the authors must perform chemogenetic manipulations of ACCp and OFCp neurons (independently) and evaluate the consequences of altered claustrum activity on sleep patterns and on the number of auditory awakening. The authors might consider testing either inhibitory or excitatory DREADDS, if possible.

Minor comments:

- Fig. 1 and Fig S2A: For consistency, use either "NREM maintained" or "Sleep maintained"

- Line 107-108: "Operationalizing NREM sleep depth via the ratio between SWA and Theta, we

observed that it shows a positive linear relationship with ACCp activity". Is the relationship absent in OFCp activity?

4- Engage task and claustrum activity

The authors developed a novel task that they called ENGAGE. Though the task is interesting, it was hard evaluating it (data are distributed everywhere in the manuscript and description of the parameters is rather limited). It is problematic since the task has never been described nor validated before. As an example, I found difficult assessing the claims made by the authors regarding the interpretation of what the task is testing. The authors wrote several times the ENGAGE task is cognitively demanding, requiring impulse control, sustained and selective attention..., but what are the data supporting some of these statements. I found the task challenging in terms of sensory perception/discrimination for sure but I have hard time estimating how cognitively demanding the task is. Most trials require discriminating between different auditory cues, signaling different information such as trial onset, go signal and the cloud. In my opinion, the task challenges different features of auditory perception: 1- discrimination between trial onset cue and go cue, 2- hearing threshold (when the go cue intensity is varied) and 3- background/foreground discrimination when the cloud is present (similar to a cocktail party problem; I don't know if I would call the cloud a distractor).

It is also difficult comparing the freely moving data to the head-fixed data since the task rules are not comparable (for example, what is the penalty for impulsive trials when mice are head-restrained? plain water was used in freely moving sessions and sucrose was used in head-fixed sessions...). It is also not clear to me why the impulse rate doesn't change during the course of learning but decreases on the head-fixed rig (Fig. S3).

The authors should spend more time describing the task and the different parameters. Right now, it is hard for the reader agreeing with the authors' claims. Alternatively, The authors should consider doing a separate study describing the characterization of the task and perform additional controls (manipulate attentional levels, use different types of distractors and/or sensory cues such as air puffs, check for ACC/AUD contribution to the behavior?...).

I have other specific comments:

- The authors claim that ACCp neurons restricts engagement but could we imagine an alternative interpretation? The authors should precisely define their definition of engagement. To me, manipulating the ACCp neuron firing with the excitatory DREADD didn't change how the mice were engaged in the task since neither the total number of trials per sessions (Fig 3E), nor success rate (Fig S7C), nor reaction time (Fig S7D) were different between saline and CNO conditions. Behavior differed regarding the temporal distribution of the different trial types during the sessions. Fig 3E shows the cumulative distribution of various trial types during the course of 3 sessions (are the distributions for Impul/Hit/Miss plotted on top of each other? If it is the case, I would suggest plotting the data using stacking histograms). CNO reduced the number of impulsive trials and concomitantly increased the hit rate at the beginning of the sessions whereas more misses are observed at the end of each sessions. How can the authors disentangle a change in engagement, from a change in attention or a change in sensory perception? If hearing threshold, auditory discrimination and/or background/foreground discrimination are improved by the DREADD, then the main effect would be an improved sensory perception and not change in cognition or engagement. Naturally, since the mice are more quickly doing many hit trials, they are more rapidly satiated (not thirsty anymore or saturated by the sucrose consumption; by the way, can the authors explain why they used sucrose instead of plain water during the head-fixed behavior - Line 147- ?) and then they wouldn't be engaged in the task anymore and therefore do mostly missed at the end of the session.

- The differences between ACCp and OFCp activity is questionable and could simply reflect a difference in sample size. For example. Fig. 2I and Fig S6B are not very different to my eyes. Doubling the sample size for the OFCp might lead to a similar significant difference. Furthermore, OFCp experiments used GCAMP6 indicator whereas ACC used primarily RGECO indicator. Since the sensitivity and signal to noise ratio of the two indicators might differ substantially, how confident the authors are that the differences are not due to the use of two indicators.

Can the authors plot the paired points for each animal?

- The chemogenetic effect is interesting but requires additional controls to convince the reader. First, I find the CNO concentration used, 10mg/kg, quite high. At such concentration, many side effects have already been reported in the literature (see for examples: 10.1016/j.neulet.2020.135432; 10.1101/2022.02.01.478652; 10.3389/fphys.2019.00521). It might be a concern since only 3 control mice were analyzed by the authors and they show the same tendency of reduced impulse rate. It would be important to do the same number of control mice and plot the data side by side with AACp. It's also very hard to compare the two groups since the authors used different normalization processes in Fig 3D and Fig S7B.

- Considering all the technical limitations and questions raised up to now, I am not sure it makes too much sense discussing the Exploiters/Explorers dichotomy in great details. However, such difference might again represent individual differences in auditory perception. How can the authors test for that?

Furthermore, have the authors used the same splitting criterion for the OFCp mice? Are OFCp explorers showing a difference?

- What would be the consequences of an inhibitory dread activation during the engage task? Would the mice only do impulsive trials?

Minor comments:

- Since both GCaMP6s and RGECO was used, I find Fig 2D drawing misleading. It should be updated.

- Line 534 "These blocks were repeated 2-4 times (as long as mice maintained participation)" how was participation evaluated? Criteria?

ENGAGE task

- How was the water restriction done?

- Is the re-initiation (leaving the port) also needed for misses?

- Since it is a novel task, an image of the apparatus might be helpful to add for the reader

- Do the mice need to reach 50-70% success before advancing to the next stage?

- Sucrose concentration and drop size should be indicated in the methods

Reviewer #3 (Remarks to the Author):

This study by Atlan and collaborators investigated the role of claustrum cells in sensory (dis)engagement in a novel cognitive task. Parallel data showing opposite correlation between sleep slow wave activities and ACC-projecting cell activity in the claustrum is proposed as a mechanism for lapses and missed trials in the cognitive task.

Both aspects of the work are very interesting and supported by studies in animal model and human showing (1) the role of the claustrum in cognitive task as well as sleep oscillations coordination across the cortex; and, (2) the importance of proper arousal, and the negative impact of slow waves occurrence during wakefulness (e.g., high sleep pressure), on cognitive performance. These lines of research have strong relevance to ageing and dementia pathologies.

Beside the writing of some of the results section that are difficult to follow but could be addressed in revision, I have several concerns on the experimental strategy and the significance of the cell dynamics reported here which would need additional data to supported the claims of the authors. First, the link between sleep SWA and disengagement is only assessed indirectly which are inconsistent with the major goal of the study (see rationale and goal in L22-28 and introduction section) and are not linked in the conclusions. This study reported changing activities of ACCp cells

across sleep states, and cognitive task in separate experiments which only provide indirect link but left the untouched the original hypothesis. The latter would require to record SWA activities and ACCp cell dynamics during the cognitive task. Without that the title and the conclusions of the study are not supported since they may be the results of a change in ACCp cell activity that is not be associated with SWA.

In other words, the implicit questions from this study remain open: Do SWA correlate with disengagement? Is the claustrum specifically implicated in this? the authors provide manipulation of claustral cell, but do those perturbations reflect awake SWA ? Potentially one may claim that any circuits in the cortex may have the same effect ? Thus, it is not clear what is specific about the claustrum here as compared to another cortical circuits. Accordingly, PFC and other input circuits are also involved in similar cognitive task...

If the goal of the study was limited to the cognitive task, without the assessment of SWA modulation of claustrum cell dynamics, then this should be clarified and the manuscript should be re-written and the data from figure 1 removed.

My second concern relate to some of the figure panels and related experimental procedures, as listed below, where the difference of activities of claustrum cell dynamics appear minimal as reported (Z-scoring value are mainly below 0.8). In this reviewer's view, it is difficult to assessed what such small changes means without having access to the raw $\Delta F/F$ values.

-Fig 1G: z-score values are quite low – how significant are those changes ? the authors must report $\Delta F/F$ values here.

The sensory disengagement and infrequent sensory-evoked awakenings is not obvious from the results shown in the figure 1. The author should clarify and justify why some animal did not follow the same number of sessions (9 of 11 for some); how many animals ? how are data analyzed? From the results presented in figure 1, the authors claimed that "These results indicate that ACCp activity mirrors sleep depth and reduced sensory engagement". This statement is correlative at best, and the evidence on sensory disengagement is weak. Simultaneous experimental measures of SWA are necessary to support this claim (see comment above and below).

-Fig 2: if I understand well, the EEG/EMG were not recorded during the experiment, so how is this assessing SWA ?

My understanding is that all trials from 20 mice are pooled together. Thus the high value of trials (>25000) is misleading and due to only few animals. Although this is informative I would suggest to separate results for each animals instead of pooling all datasets. In addition, there are as many hit as missed trials? Was this expected ? if animals learned the task, one would expect a decrease of miss, isn't ?

-Fig 2E-I: Z-scores of data are quite low suggesting minimal differences between experimental conditions. Please present raw data in parallel and explain the relevance of such low Z-scores.

Regarding the task itself, Why tone-cloud distractor and visual stimuli are presented in some of the trials and not all of them? It is unclear how these are affecting performance as data do not systematically take into account these 'co-occurring' stimuli. Do they have any effect on the performance ? or ACCp activation, etc? please clarify.

-Fig 3B: Does CNO/DREADDS activation of ACCp reflect activity similar to SWA or sleep ? it seems to induce massive waves of activity with a very long periodicity (± 20 s ?) much slower than previously reported SWA – what does that means ? Are those similar to natural/spontaneous activity ? The effect on the performance is clear, however, it is difficult to conclude that this is due to SWA.

Note that proper control for CNO experiments is actually CNO injection in DREADD-negative animals since the compound itself may have effect on brain dynamics and behaviors.

-L157: why were animals tested in a head-restrained set up while training was in freely-behaving context ? This may have major influence on performance, SWA etc. and I don't see how this is being controlled or tested for in the data set. Please clarify.

Minor concerns:

L76: ACC-projecting neurons in the claustrum (ACCp) is confusing – the authors should use CIACC or other alternative names but this should be clarified.

L89: “reliance” – please reword.

Fig1A: “R” form retro-AAV is missing

Fig 1C: color coding of dots is difficult to see – please adapt.

Fig 1E: could the authors provide quantification and statistics for those those graph ?

Claustrum Neurons Projecting to the Anterior Cingulate Restrict Engagement During Sleep and Behavior Response to Reviewer Feedback

Summary of major changes:

New experimental data:

1. We performed a new experiment, expressing the DREADD hM3Dq in ACCp neurons, the results of which demonstrate the causal relationship between ACCp activity and NREM sleep, as well as the causal effect of elevated ACCp activity on sensory disengagement during NREM sleep (Figures 1 & S1).
2. We performed additional control experiments to rule out non-specific effects of CNO on behavior and ACCp activity during the ENGAGE task (Figures 3 & S5).
3. We added fiber photometry experiments illustrating the distinction between ACCp and OFCp circuits (Figure S2).
4. We performed experiments directly addressing the correlation between axonal and somatic activity of ACCp neurons recorded via fiber photometry (Figure R5, below).

New analyses:

1. We have significantly revised the analysis and presentation of the experiments addressing the relation between ACCp activity and NREM sleep, as well as ACCp activity and sensory-evoked awakening (Figure 1 & S1).
2. We have added analyses addressing the long-term stability of ACCp recordings during sleep (Figure S1).
3. We provide additional analysis of the physiological impact of chemogenetic stimulation on calcium activity in ACCp neurons, illustrating a specific effect on event amplitude. We also provide a more detailed exploration of ACCp-Gq activation on trial outcomes and behavioral strategy, illustrating the specificity of the effect of CNO and the selectivity of its impact, reducing impulse errors (Figures 3 & S5).
4. We provide more detailed presentations of the data for individual mice throughout the manuscript (Figures 1, 2, 3, 4, S1, S2, S4, S5).

Graphical and textual changes:

1. The reviewers identified several aspects of the data and presentation which were not directly related to the main thrust of the paper, and generated potential confusion for the reader. We have removed these elements while ensuring that essential components were retained. These include some of the analyses relating to the relationship of the ACCp and OFCp populations (formerly figure S1), the training data prior to recordings in the ENGAGE task (formerly figure S3) and some of the sleep analyses (formerly figures 1 & S2).
2. Based on the reviewers' comments, we have revised and simplified the abstract, introduction, results, discussion, and figure legends, improving clarity and accessibility.
3. We have embedded all statistical information within the figure legends, increasing transparency and accessibility.

Point-by-point response to reviewer comments:

Original comments in *black italics*. Our responses in blue.

Reviewer #1: *The authors perform calcium fiber photometry in the claustrum in a novel sensory response task. Interestingly, they find that the claustrum is most active during lapses in engagement, during trial misses. **This is a very important result. The paper is timely** as another recent study was published last year showing claustrum neuron modulation on a multimodal sensory detection task. This new paper provides complimentary data and extends the new idea that claustrum may be involved in disengagement (an idea that is controversial, but nonetheless supported by the data). **I have some comments on the analysis, and writing, but otherwise the paper is very interesting.***

We thank the reviewer for their interest and support.

#1: In Figure 1, is it the case that slow wave activity is also decreased prior to the awakening events? The impression is given that the reduced claustrum activity is 'causal' in the awakening, but perhaps the state of the mouse is generally different and claustrum activity simply covaries with activity in the cortex? Perhaps there is a way to determine if the variability in claustrum activity can account for the differences in awakening versus not (above and beyond that information provided by slow wave activity). Or maybe these 2 variables are too intimately linked, or too few trials to be able to conduct the analysis. In any case some further analysis or discussion of the interpretation is warranted.

We thank the reviewer for highlighting these important issues. We addressed this via several complementary approaches including a new experiment testing for causality (Figure 1F-L), as well as analysis aimed at separating the contribution of claustrum activity from slow wave activity (SWA).

First, in order to directly probe the relationship between ACCp activity and awakening, we experimentally elevated ACCp activity while measuring tone-evoked awakening. This experiment demonstrates a causal role for elevated ACCp activity in limiting sensory-evoked awakening (Figures 1J-L, S1H-J).

Second, to determine if the variability in claustrum activity can account for the differences in awakening - above and beyond that information provided by SWA - we applied a linear mixed model (LME) to predict the probability of awakening, addressing the contributions of either the ACCp signal alone, SWA activity alone, or the combination of the two signals as predictors. The three models were all similarly predictive, and a likelihood ratio test was not significant. The co-linearity between ACCp and SWA was not significant (Pearsons R=0.04, p=0.14). This implies that these two predictor variables (ACCp activity and SWA) are independent, and while each contribute to the waking probability, they do not represent redundant information.

Third, it should be noted that the relation of ACCp activity to sensory-evoked awakening is not a brain-wide phenomena, as it is not shared with the neighboring population of OFCp neurons (Figure S1F,G)

Fourth, following this comment, as well as the comments of the other referees, we realized more generally that the revised manuscript would benefit from clarifying its focus on the relationship between ACCp activity and engagement, and decided to remove the emphasis of the original manuscript on SWA. Instead, we address the relationship with SWA in the 3rd and 4th paragraphs of the discussion, as a potential avenue for further study of the mechanism whereby ACCp neurons control engagement.

#2: I do not really understand the significance of the main effect of mouse identity in Figure 1E. The last sentence in paragraph 1 of the results mentions a positive linear correlation with respect to the quartiles of claustrum activity, but I do not see this reported in the figure legend or text.

As the reviewer points out, the description of this panel could have been clearer. We mentioned mice identity to explain that individual subjects were treated as random effects (data points) in the paired comparison, a suboptimal wording to describe the statistical approach. As explained above, in the revised manuscript the relation between SWA and ACCp activity is no longer emphasized, so this result (former panel 1E) has been removed from the revised manuscript.

#3: What are the units for Figure 3C on the y axis? In Figure 3, I find it odd that the signal fluctuations are compared between the 2 treatments rather than measuring how much each treatment impacted the signal independently, relative to the time period before each injection (as signals can vary across days even in a single mouse). For example, why not compare baseline versus CNO in 1 session, then baseline versus saline in another session (then compare the change from baseline between saline and CNO).

We have replaced Figure 3C with an improved panel, in which we plot the impact of CNO on the ACCp signal (z-scored DF/F) as a modulation index $[(\text{CNO}-\text{saline})/(\text{CNO} + \text{saline})]$; limited between -1 and +1], and include a comparison to a larger cohort of control mice (Figures S5F,G).

It is important to point out, that in the experiments performed in the context of the ENGAGE paradigm, we injected mice in their home cage (i.p., CNO vs saline) and connected them to the recording setup 15-30 minutes after the injection. This approach was used since in initial experiments we found that any intraperitoneal injection performed on the rig while the mice were head-restrained induced significant stress. Therefore, the signal is compared across recording sessions. We have improved this description in the methods section.

Lastly, in order to address the variability across experimental days as requested by the reviewer, we conducted a new analysis, plotting the average ACCp signal for each mouse on each day of either CNO or saline exposure. The data is baseline subtracted based on the first saline day and illustrates the robust and reproducible effect of CNO, elevating ACCp activity across all experimental days (Figure R1).

#4: In Figure 3 the relative error rate is shown for CNO and saline. The critical control experiment is shown in Figure S7, but in this case the y-axis is labelled impulsive error rate. Why not show the same measurement between the experiment and control? Also, in Figure S7, perhaps the trial type can be

broken down into hits, misses, and impulsive. The dose of CNO is very high, so these controls are important here. I'm not sure the 3 control mice are particularly compelling given that it would be hard to show any significant difference with and $n=3$.

We now represent the effects of CNO on control mice in the same way as for experimental mice, by normalizing their impulsive error rates following saline or CNO to those in baseline sessions that preceded exposure to saline and CNO (Figure R2). As requested, we have performed additional experiments, increasing the number of control mice to $n=5$ (Figures S5F,G).

In addition, as suggested, we also include the analysis of the raw effect of CNO on all trial outcomes (impulsive, correct, miss), in both experimental and control animals (Figures 3E, 3F, S5B, S5G).

#5: In Figure 3 the authors can provide further support of their hypothesis, because they are measuring ACCp activity during the behavior. I would recommend determining how ACCp activity is organized on hit, miss, and impulsive trials with hm3d-CNO, as this would provide further evidence of the role of these neurons in task performance.

This is an interesting proposition. Following the reviewers' comments, we have further explored the impact of CNO on the ACCp photometry signal. By far, the most prominent effect is the elevation of baseline activity (Figures 3B, 3C, S5A, see also Figure R1), consistent with the hypothesis regarding the association of elevated pre-trial activity with reduced probability to perform impulse errors, and increased probability to perform trial omissions. This drastic shift in baseline activity precludes the interpretation of other, more modest shifts in activity.

#6: Related to the above point, when discussing the CNO results, the authors say that CNO did not impact task associated responses in the ACCp signal, but I cannot see any statistics to support this (Figure S7E). Also, with such a low sample size, I suspect detecting significance between these conditions would be challenging. Consider restating this in the results if statistical comparisons cannot really be drawn here.

The statistics for former Figure S7E were reported in the supplementary statistics table as paired t -tests for each event label. While the variance in the response magnitude appeared to increase, we did not observe any significant changes in evoked responses under CNO, even before correcting for multiple comparisons. Unlike the pre-trial activity, evoked transient responses in the ACCp signal showed little contribution to behavior in the ENGAGE task. Given these findings, we speculate that they reflect incoming inputs to ACCp neurons. Supporting this are data from cortical axons in the claustrum, showing similar transients to claustral activity recorded simultaneously during the task. We reported these

observations in the preprint of this manuscript, posted online (Atlan et al., 2021). However, as detailing this interesting yet relatively minor point mandates a significant excursion from the thrust of the manuscript to present and explain, we believe that this observation is outside the scope of this study, and have therefore removed the analysis of evoked transients under CNO from the revised version of the manuscript.

#7: Is there any indication if the 5 mice receiving hm3d are exploiters or explorers according to the criteria in Figure 4A?

We thank the reviewer for this interesting question. We analyzed the behavior of the DREADD mice, plotting them on the same axes of impulse error rate vs. cloud modulation, as used for defining the behavioral categories in Figure 4A. All 5 hm3Dq-expressing mice fall initially within the category of ‘explorer’ mice. Following CNO, all mice exhibit a dramatic and selective reduction in impulsivity. We have included this analysis in the revised version of the manuscript (Figure 3G). The shift in performance induced by CNO shifts the mice from ‘explorers’ to the region in the graph inhabited by ‘exploiters’. However, as the effect of CNO is selective to impulse errors, with no impact on visual modulation or cloud modulation, we hesitate to characterize it as a shift in strategy. More importantly, the shift in performance is transient, and reverts back in the absence of CNO, consistent with the impact of Gq-induced elevation of ACCp activity eliciting a focal impact on task performance, rather than driving a transition between behavioral strategies.

#8: In lines 283-285 the authors state that a simplified response strategy may lead to independence from the claustrum implying that this difficult task is somehow recruiting the claustrum. When in fact their data are leading me to believe that claustrum activity does not help the mouse in this task very much at all. Increased claustrum activity in explorers seems to correlate with misses, whereas effective performance (assessed by hits) may or may not correlate with moderate levels of claustrum activity (the authors do not correlate claustrum activity with hits). The word “recruited” implies that claustrum activity is playing a role in performing the task, but again, I do not see the evidence for this. Please help me as a reader overcome this confusion, or please clarify the language. Perhaps my confusion is rooted in the fact that they state that exploiters use a simplified response strategy (and do not have different claustrum levels pre-trial), but how can the authors make this claim regarding what strategy is simple versus hard? It seems both strategies have their merit.

We thank the reviewer for this important critique. Our intention in using the term ‘recruited’ was to imply a relationship between claustrum activity and performance. A relationship which is stronger in *explorer* mice than in *exploiter* mice. However, this is indeed confusing, as the reviewer states. In light of this comment, we have completely revised the writing of both the results and discussion sections of the manuscript, removing discussion from the results section and clarifying the intention of the interpretation in the discussion. As the reviewer mentions, both strategies have their merits. We have added text to explain why we believe the ‘exploiter’ strategy simplifies the task by greatly reducing the need to inhibit responses at trial onset.

#9: In the discussion, the authors state “our results demonstrate a causal role for pre-trial claustrum activity...”. This is imprecise. The causal experiment does not analyze pre-trial activity. How do we (as readers) know that the hm3d manipulation can be explained by the effects on pre-trial activity at all, or specifically pre-trial activity. Sentences like this, start to imply that perhaps loss of function experiments or temporally specific (optogenetic) experiments should be done. I suspect the authors do not want to add these experiments to this paper. So, I would recommend limiting sentences that overstate the reach of the results.

We thank the reviewer for another important and constructive critique. We have edited the discussion to better align with what can be safely inferred from the data.

#10: Another part of the discussion mentions the decrease in impulsive behavior with the hm3d (Line 313). However, the authors do not mention the prominent increase in misses with hm3d-cno. These hm3d mice are not doing particularly better on the task (if anything, their hit rate goes down). Perhaps this should be highlighted as well in the discussion, as this is also a major finding in the paper (the correlation between increased claustrum activity and misses).

We fully agree with the reviewer, and have included a more detailed description of the effects of claustrum activation on task omissions. We have also included the graphs documenting the hit and omission rates following CNO (Figures 3E, S5B, S5F, S5G). Overall, misses do increase with CNO. However, when compared daily, taking into consideration the effects for the individual mice, the effect was not statistically significant. Hit rates were largely maintained under CNO.

Reviewer #2: *In this study, Atlan and colleagues tested the hypothesis that the claustrum may regulate engagement. They recorded different populations of claustrum neurons either projecting to the anterior cingulate cortex (ACCp) or to the orbitofrontal cortex (OFCp). They suggest that ACCp neurons, but not OFCp ones, are more active during sensory disengagement during sleep. In addition, using a novel behavioral assay, the authors showed ACCp activity was high when mice were hypo-engaged and prone to behavioral lapses and was low when mice were doing impulsive responses. Chemogenetic elevation of ACCp activity suppressed impulsive behavior and reduced task engagement.*

*I have a mixed feeling about the study. On one hand, the hypothesis tested and some of the data are interesting. On the other hand the paper is difficult to read (mostly because it's not well structured, important information being scattered all over the place) and suffers from missing experiments/controls, possible alternative interpretation of the data and technical limitations. In its present form, the paper is in my opinion not acceptable. The authors will need to revise the study in depth, answer to the different comments and rewrite the manuscript (expand the text, reorganize and increase the number of main figures) before being published. However, **if the authors address the criticisms, I have no doubt the paper will be important for the neuroscience community at large and would deserve publication in Nature Communications.***

We accept the reviewer's critique that the paper was complex and difficult to read in the previous version, and have performed a thorough revision, simplifying and restructuring the data presentation and the writing, as well as adding new experiments and analyses, focusing on presenting the data essential for supporting the main findings of the study. We thank the reviewer for prompting us to undertake this major revision and are confident that the revised manuscript is substantially improved.

See specific comments below:

1- Characterization of different groups of claustrum projection neurons

The entire study relies on the analysis of claustrum neurons, labelled with retroAAV-cre, specifically projecting to either ACC or OFC areas. I have no doubt that the authors are claustrum experts, but I am quite confused by the way claustrum is defined in this study and by the quantification done. Although the authors showed some characterization of the labelled populations in Fig. S1, it is clearly not enough to ascribe cells to the claustrum and not the insular cortex. Undoubtedly, a much better characterization of the labeled populations is needed to interpret the results and support the claims of the study. See specific comments below.

Comments #1-14 of reviewer #2 all relate to Figure S1 (current Figure S2), which deals with comparison of the ACCp to OFCp neurons. Taking all these comments into consideration, we have thoroughly revised this supplementary figure. Before addressing each comment individually, we will outline the essence of our response and the changes we have implemented.

The focus of our study is on the population of claustrum neurons projecting to the anterior cingulate cortex (ACCp). Claustral neurons projecting to the ACC are the most extensively studied cell population in the claustrum. They have essentially become the consensus reference population for claustrum projection neurons, recognized by the majority of claustrum research groups (Chia et al., 2020; Erwin et al., 2021; Jackson et al., 2018; Marriott et al., 2021; Narikiyo et al., 2020; Ntamati et al., 2023; Peng et al., 2021; Qadir et al., 2018; Reser et al., 2017; Smith et al., 2019; Terem et al., 2020, 2023; Wang et al., 2023; White et al., 2017, 2018; White & Mathur, 2018a, 2018b). This is due to the fact that a consistent cluster of neurons is targeted with retrograde labeling from the anterior cingulate cortex, with very little labeling of neurons extending beyond the region labeled in brain atlases as defining the claustrum (Franklin & Paxinos Atlas and Allen Brain Atlas). It is important to note that the OFCp population, while analyzed in detail, is included as a secondary comparison (in supplementary figures).

Relating to the comparison of ACCp and OFCp populations, we have performed a thorough restructuring of former figure S1 (now Figure S2). We have added a characterization of the spontaneous activity in ACCp vs OFCp networks (Figures S2E-K) recorded by fiber photometry, comparing recordings across mice (Figures S2E-G) as well as co-recorded utilizing dual-color simultaneous photometry (Figures S2H-K). We have also simplified the presentation of the anatomical characterization, removing the reference to core and shell structures in the claustrum and the quantification of the axonal projections of ACCp and OFCp populations. The comparisons included in Figure S2 now clearly demonstrate that while these networks do exhibit some correlation and overlap, they are largely differentiated both in their spatial distribution, as well as in the characteristics of their population activity, as reported by calcium fiber photometry.

#1: - From what I can see in the Allen brain atlas, the claustrum is described as a small oval island (of ~300-600 microns in diameter) in the agranular insular cortex. Therefore, claustrum size and shape used by the authors seem to be quite different from the one defined by the Allen Institute. As an example, H2B-tdTomato labelled neurons are distributed over more than 2mm in the DV axis in Fig. S1B, which seems too much to be only the claustrum. How can the authors exclude the targeting of insular neurons? For example, some insular neurons display partially comparable projection patterns (see https://connectivity.brain-map.org/projection/experiment/siv/296048512?imageId=296048530&imageType=TWO_PHOTON,SEGMENTATION&initImage=TWO_PHOTON&x=17806&y=11081&z=3; or https://connectivity.brain-map.org/projection/experiment/siv/485847695?imageId=485847813&imageType=TWO_PHOTON,SEGMENTATION&initImage=TWO_PHOTON&x=16509&y=14985&z=3; https://connectivity.brain-map.org/projection/experiment/siv/166153483?imageId=166153633&imageType=TWO_PHOTON,SEGMENTATION&initImage=TWO_PHOTON&x=24672&y=15052&z=3).

We thank the reviewer for identifying a typo in the scale bar included in figure S1B of the original submission. Indeed, the medial-lateral axis of the claustrum is on the order of magnitude of 300-500 microns. While there is a slight discrepancy between the definition of the claustrum in the Allen Brain Atlas and the classic Franklin & Paxinos Atlas, the population of ACCp neurons falls clearly and consistently within the region that is a consensus across both these atlases, as well as many published studies of the claustrum. The populations we label in our work are a fraction of the size of the examples cited by the reviewer, which encompass not only the anterior insula but also the claustrum and, at times, extend well beyond these regions. We are careful to calibrate virus titer and injection volume for every batch virus we use, so as to obtain expression that is restricted to the claustrum.

#2: The authors could probably use antibodies against known claustrum and cortical markers in order to characterize the different populations targeted by the two retrograde injections. I also noticed that the authors used in a previous publication a transgenic line with specific cre expression in claustrum neurons. How comparable are the cells labelled with the retroAAV and the ones found in the mouse line?

We assume that the reviewer refers to our work with the Egr2-Cre mouse line (Atlan et al., 2018). As described in that publication, Egr2-Cre labeled claustrum neurons project to both the ACC and the OFC, as well as to other targets. Thus, while we have not performed a direct comparison of ACCp neurons to the Egr2-Cre defined neurons in the claustrum, we deem it likely that the EGR2⁺ population captures more than a single projection network. In recent years, we and others in the field have shifted our work to focus on claustral neurons captured by their connectivity – a strategy we believe to be more productive in uncovering neuronal function, as we stress in the revised discussion. We find the definition of claustral projection neurons by the identity of their projection target to be the most unambiguous approach of defining claustral neurons, with distinct benefits over identification based on the expression of genetic markers. The expression of genetic markers may vary based on activity, genetic background and other variables. Furthermore, genetic markers provide limited information about the identity of the neurons they capture, beyond the name of the gene they express. In contrast, defining neurons by their projection segregates populations based on anatomy (by definition) and is more likely to be stable across conditions, mouse lines and between labs. We, and others, have addressed the identity of frontal-projecting claustral neurons in prior publications (Chia et al., 2020; Erwin et al., 2021; Jackson et al., 2018; Marriott et al., 2021; Narikiyo et al., 2020; Ntamati et al., 2023; Peng et al., 2021; Qadir et al., 2018; Reser et al., 2017; Smith et al., 2019; Terem et al., 2020, 2023; Wang et al., 2023; White et al., 2017, 2018; White & Mathur, 2018b). Creating a repository of the overlap between genetic and anatomic definitions of claustrum neurons would definitely be a worthy undertaking for our community, but is outside the scope of the current study. Additionally, identifying a specific function for a population of projection-defined neurons, as we do here, provides a reference point for future studies of genetically-defined populations.

#3: - In line with the previous comment, can the authors provide a clear definition of what they consider claustrum core and shell. From what I could read in the literature, it seems that PV staining is usually used in this field to distinguish the two. However, on the image shown on Fig S1D, I noticed ACCp neurons are found mainly outside the core, though it might be a problem of image quality in the pdf file. Similarly, the magenta signal seems to surround a core at bregma 1.1 and -0.7 in Fig. S1F.

The reference to core vs shell regions of the claustrum is immaterial to the current form of our manuscript ('core' vs 'shell' only appeared previously in the legend to former supplementary figure 1) and was therefore removed.

#4: - There seems to be much more OFCp than ACCp cells (Fig. S1B,C). However, the density plots give a different view. Unless I missed the information, I couldn't find any quantification regarding the number of labeled neurons in the two experimental conditions. The authors should plot this information at the different bregma levels since this might impact the interpretation of the imaging/behavior results (see below). The authors should also explain how the densities shown in Fig. S1C,E were computed.

The reviewer is correct in noting there are roughly double the number of OFCp neurons than ACCp neurons. The density plot was indeed normalized in order to allow visualization of the spatial distribution, so differences in cell number were not emphasized. We provide the information here for the reviewer (Figure R3), but do not find it to be of sufficiently broad interest to merit inclusion in the current manuscript.

Figure R3: Proportion of ACC- vs OFC- projecting claustral neurons along the rostral-caudal axis of the claustrum. AAV-retro-H2B viruses colored with either GFP or tdTomato fluorophores were injected into the ACC & OFC (n=7 mice).

#5: - The authors used the spatial distribution of the two populations to make the claim that they are different. It might be true but the data are not compelling especially since the fraction of double labeled neurons remains relatively small when injections are made at the same coordinates (=same cortical area, for example 15% for ACC using nuclear-targeted fluorescent proteins which is less ambiguous, top panel Fig S1G,). To convince the reader, in addition to the characterization mentioned above, the authors should plot the same distributions displayed in Fig S1C,E for the experiments where double ACC or double OFC injections were made. How do these distributions compare to the ones already reported in Fig S1?

The double labeling data was meant to address the limitations in co-infecting single cells with multiple retrograde viruses. It is indeed hard to make definitive quantitative statements using this method given the data we have available. For the sake of simplicity, we have removed the analysis of double labeling and kept only the spatial distributions to demonstrate their different spread in the claustrum. In addition, we have added analysis of the activity of ACCp vs OFCp populations, as measured by fiber photometry, comparing the same calcium indicator across mice (Figures S2E-G) as well as dual-color simultaneous recordings within the same mouse (Figures S2H-K). These results demonstrate the extent of correlation between ACCp and OFCp populations, as well as the similarities and differences in the physiological activity of these populations.

#6: - In line with the previous comment, the authors used the axonal spread of the two populations to make the claim that they are different (Fig S1H,I). Are the axonal distribution really different (please plot individual data points and provide statistical tests)? I agree that ACCp mice show more axons in ACC than OFCp mice (and vice-versa). However, were the same quantification done for experiments with double injections in the same cortex? How variable are these results?

As requested, we have plotted the individual data (Figure R4). To simplify the presentation in the revised manuscript and avoid overstating, we chose to exclude the quantification and keep only the representative histological images.

We have not studied axonal projections by expressing two fluorophores in the same hemisphere, so we cannot provide insights in response to this comment. One of the reasons we did not do so is that this experiment, by definition, would make use of viruses expressing different fluorophores, adding additional experimental variation and complexity to the analysis.

Figure R4: Data from individual mice (ACCp=blue, n = 4; OFCp=green, n = 3) for comparison of axonal projections of projection-based targeted claustrum neurons. Y axis reflects the average 8-bit pixel intensity (0-255) in each structure. Differences between the two groups of mice were observed in the target regions (i.e. ACC and OFC), as well as in posterior sensory cortices (i.e. AUD, RSP, and VIS). The limited number of animals prevented meaningful statistical testing.

#7: In addition, I did not understand the way the axonal quantification was done. In Line 478-483, you wrote “a manual threshold was set for every brain such that the claustrum area would be saturated and background would be minimal, enabling a clear contrast for the fluorescent processes. Structures of interest were selected based on previous anterograde tracing studies in the claustrum (27). Analysis was conducted in Fiji (ImageJ) and quantified as the mean pixel intensity in a rectangle sampled within different brain divisions. Measurements were obtained from qualitatively similar positions in each section over mice”. How was the threshold decided and how much is this changing the results if varied? How can different brain slices be analyzed? What is the background level? Finally, are we looking at average pixel intensity in each area analyzed or have the authors used any form of normalization?

The analysis calculated the average pixel intensity across rectangles overlaid on different brain regions. No thresholding was performed, and the only normalization was adjusting the brightness of the image so as to obtain saturation (pixel intensity = 255) of the region of transfected cell bodies in the claustrum in each section. To address the reviewer’s comment and avoid the inherent complexities in quantification of such images, in the revised manuscript we only present representative histological images to demonstrate the difference in the spread of projections.

Minor comments:

#8: - Fig S1C left panel: Could the authors plot a graph with actual distances instead of a.u.? What is the reason for using a.u.?

The histograms included in revised figure 2B are accompanied by scale bars (representing 100um) to demonstrate the actual distances.

#9: - *Can the authors indicate which PV antibody was used for histology. Was the alexa647 (Line 452) used for this staining?*

We used Sheep anti-PV from Biotechne (catalog #AF5058), with donkey anti-sheep Alexa Fluor 647 from Enco catalog #713-605-003). As this panel has been removed, this information is not included in the revised version of the manuscript.

#10: - *Fig S1E the colors used to label the different cells are too similar to be discriminated even on a computer screen. I suggest using other colors (why not green and magenta?)*

Prior to publication, we will gladly work with the journal to ensure that color discrimination is satisfactory and improve the visualization further if necessary.

#11: - *Can the authors specify which experiment shown in Fig S1 use a PV-cre mouse (as mentioned in Suppl. Table 1).*

Thank you for capturing this error. This is a relic from a prior version of the manuscript, and has been removed in revision.

#12: - *Fig S1I: AOB should be changed for AON*

We removed this panel from the manuscript.

#13: - *What kind of image acquisition system was used? Confocal or epifluorescence microscope?*

We used epifluorescent microscopy. This has been clarified in the methods section of the revised manuscript.

#14: - *Fig. S1I: Have the authors excluded prelimbic cortex from the analysis or combined it with rACC? The authors should also define in the methods how rACC, mACC, cACC, CVO, rVO were defined.*

rACC included prelimbic cortex. We have removed this panel from the manuscript.

2- Calcium imaging using fiber photometry

The authors used photometry to specifically monitor activity from ACCp or OFCp neurons. Since photometry records an averaged population signal, it is likely impacted by the number/distribution of labeled neurons, and might not even necessarily report firing activity (see below). Since the authors claim that ACCp and OFCp neurons behave differently, important questions still need to be addressed before trusting the calcium data. See specific comments below:

#15: - *The authors used calcium photometry signals as a proxy for firing activity. However, a recent report convincingly showed that photometry don't primarily reflects somatic changes in calcium, at least in the striatum (see the publication DOI: 10.1038/s41593-022-01152-z). Since the entire study correlates claustrum "activity" to behavior, it is important to understand the source of the recorded signals (spiking, neuropil, dendritic information...). Can the authors estimate the number of imaged soma? Can the*

authors exclude that differences between ACCp and OFCp response represent differences in nonsomatic calcium responses (for example, ACCp may differ in their dendritic and/or axonal arbors)? Could the authors backup some of their claims with miniscope imaging or extracellular recordings?

We are aware of the recent publication cited by the reviewer (Legaria et al., 2022). We and many other colleagues have reservations regarding its experimental design, analytical approach, and the extent to which their conclusions can be generalized to other brain regions. We have also recently published evidence refuting some of the conclusions of this publication (Lipton et al., 2023).

In any case, it is important to note that the raised issues stem mainly from the morphology of striatal medium spiny neurons (essentially spheres), which is vastly different from claustral neurons (large and diffuse dendritic trees, very long axons extending throughout the brain).

Furthermore, we are not sure that 1-photon miniscopes are more likely to eliminate the contribution of neuropil, especially since this is a challenge even for multiphoton imaging (Grienberger et al., 2012; Tischbirek et al., 2017).

Nevertheless, to directly address the reviewer's question, we have performed concurrent recordings of both the soma region, as well as the axons of ACC-projecting claustral neurons, placing an optic fiber above the claustrum, and another above the ACC. We observe correlation of axonal activity with the activity of claustrum ACCp somata, suggesting that somatic ACCp activity, as measured throughout our study, reliably reflects information that is transferred from the soma to the axons, and is not dominated by neuropil activity (Figure R5). It should be noted that the axons from the claustrum bifurcate, spanning multiple square millimeters in the ACC, limiting the capacity of a single fiber to optimally capture the fluorescence of the whole axonal projection field. This technical limitation forces an under-sampling of the activity in axons. Therefore, the correlations identified with this approach likely under-represent the true correlation in activity between soma and axons. In the case of the ACCp population, in which the soma are relatively tightly packed, while projections fan out broadly in cortex, this underestimate is expected to be more severe than in other populations (e.g. striatal neurons projecting to the GPe, as in (Lipton et al., 2023)).

Figure R5: Simultaneous calcium fiber photometry from soma and axons of ACCp neurons. (left) Representative traces; (right) correlation (pearsons r). GCamp6s expression was targeted to ACCp claustral neurons, by retroAAV-CRE injection to ACC and DIO-GCamp6s to the claustrum. Fibers were placed above the claustrum and the ACC. Activity was simultaneously measured from ACCp soma and axons (n=4 mice).

#16: - In line with previous comments, how does the number and the distribution of labeled neurons, as well as the optic fiber position, influence the photometry conclusions? For example, considering that more OFCp neurons seem to be labelled with retroAAVs and since photometry represent an averaged population signal, it is possible that specific neural responses from a subpopulation (especially if insular

cortex and claustrum neurons would be simultaneously targeted, see section 1) are down in OFCp signals but less in ACCp signals. As a second example, it seems that the authors imaged activity from caudal claustrum where there are less cells (RostroCaudal 0 from Suppl. Table 1 and as seen by the few labeled neurons in Fig. S7A). Would the results be different for more rostral location where there are more neurons? It is important to have such information since chemogenetic experiments targeted also frontal claustrum (RostroCaudal +1). Finally, as seen in Fig. S7A, the number of labelled neurons can be quite low and the position of the optic fiber lateral to the recorded field, is this kind of recording giving calcium signals?

By and large, the answers to the questions presented by the reviewer are addressed by the data already presented in the revised manuscript. With some variability across mice, we observe consistent representation of major task parameters (e.g. cue and lick responses) in the ACCp (& OFCp) responses. Distinctions that we do observe between mice (e.g. differential pre-trial activity and differential representation of the trial onset BBN) are correlated with different behavioral strategies of mice. Therefore, any variance in cell number or fiber location across mice does not appear to be the major determinant of differences we report in the association of neural signal to behavior. While it indeed may be of interest to record and compare the activity of claustral populations across the rostral-caudal axis of the claustrum, this is beyond the scope of the current study.

Importantly, the thrust of the manuscript is the recording of the activity from ACCp neurons, which are consistently devoid of contamination of neurons located outside the claustrum. Our results (from 20 mice) are consistent across individual animals, suggesting that any issue of cell number is not a major determinant of the signal quality. It should also be noted that many of the parameters of the OFCp signal are largely similar to the ACCp signal, such as the average event amplitude, dF/F MAD (Figure S2E), representation of sensory events, locomotion and licks (Figure S4). The ACCp and OFCp populations differ in the relation of pretrial activity to sensory-evoked awakening (Figure S1) and trial outcome in the ENGAGE task (Figure S4), as well as in the rate and width of spontaneous events (Figure S2E). While the OFCp may, in principle, include a modest contribution of insular neurons, in our cumulative experience over years of intersecting retrogradely transporting CRE-expressing viruses localized to the OFC, with injection of a small volume of a CRE-dependent reporter to the claustrum, the retrolabeling of OFC-projecting neurons from outside the claustrum is largely negligible. Specifically in the current study, the vast majority of OFCp neurons are found in the claustrum, over which the fiber is localized, and therefore the potential contribution of the small proportion of potentially captured insula neurons, which are peripheral to the light path, is expected to be quite modest. The reviewer relates specifically to former panel S7A, in which there are indeed a small number of neurons captured. These neurons are clearly within the boundaries of the claustrum, and are directly localized under the fiber. Our experience is that this system is sensitive enough to reliably report the activity of a small population of neurons, allowing us the freedom to perform small volume focal injections of viruses to the claustrum, and retain a high probability of success in specifically capturing our population of interest.

#17: - I found all the experimental configurations used in this study very confusing. There are multiple calcium indicators used (GCaMP6s and RGECO1) and neuronal targeting was done using either flp or cre-dependent viruses. Photometry recorded either a single signal or multiple signals at once (including ACC, OFC and even auditory cortex, and input/output dual imaging). I acknowledge that the authors honestly reported all the different experimental variations in Supplementary table 1. However, most of these conditions are neither discussed nor used in the manuscript. Furthermore it's nearly impossible evaluating the possible confounding factors introduced by such diversity and the low experiment numbers for some configurations. Can the authors demonstrate the reproducibility of their findings irrespective of the configuration/indicators used.

We thank the reviewer for highlighting the complexity of experimental configurations. Through this feedback, we realized that our aim for transparency and completeness came at the expense of clarity and focus. Following this input, we have greatly simplified presentation in the revised manuscript.

Accordingly, now only ACCp and OFCp channels were included, so as not to distract from the main focus of the study. We are confident that our main results and conclusions are not impacted by these variations in configuration. As an example, we illustrate the correlation of pre-trial ACCp activity with impulse errors as recorded using the two configurations (Figure R6).

#18: Can the authors also show examples and quantification for the experiments in which ACCp and OFCp were simultaneously recorded. Are differences between AACp and OFCp activity still observed in such cases?

The 5 mice in which the ACCp and OFCp populations were recorded simultaneously happen to be largely 'exploiters', i.e. mice that exhibit reduced impulsivity, a greater reliance on the prominent cues, and diminished impact of the tone cloud mask (Figure R7). The signal of these mice also exhibited a diminished representation of the 'no go' BBN played on trial onset, as well as a reduced relationship of pre-trial activity to trial outcome, the two main differences we observed between ACCp and OFCp neurons. By and large, with relation to the ENGAGE task, the ACCp signal of these mice appeared more similar to the signal of the OFCp mice. However, the global correlation and cross-correlation between the ACCp and OFCp signals within these mice was quite modest (± 0.3 ; Figures S2J,K), suggesting that the ACCp and OFCp signals reflect the activity of largely distinct populations of neurons.

Figure R6: Comparison of the relation of ACCp signal to impulsive errors across mice expressing either the jrGECO or GCaMP6s in ACCp neurons.

Figure R7: Mice in which ACCp and OFCp signals were measured simultaneously are 'exploiters'. (left) The simultaneously recorded jrGECO-ACCp and GCaMP6s-OFCp mice fall within the category of 'exploiter' mice (n=5 mice). (right) Correlations of the ACCp and OFCp signals within the context of the ENGAGE task, exhibiting modest overall correlation.

#19: Was the dual GCaMP6s and RGECO1 imaging truly simultaneous? How was the motion artefact corrected in this case?

As described in the methods, the recording was performed truly simultaneously, with each wavelength riding on a uniquely modulated sinusoidal wave that allows on-line de-modulation. As stated in the methods, we evaluated the potential for motion artefacts from the isosbestic signal in single-opsin recordings, and observed it to be largely negligible (an advantage of the head-fixed configuration). We observed a potential motion artifact in 2 ACCp mice, for which we applied correction based on the isosbestic signal (these mice only had a GCAMP channel and no jrGeco channel).

#20: - Can the authors exclude that photobleaching and/or phototoxicity play a role in the interpretation

of the results for long recordings? For example, is the relationship between claustrum activity and either sleep or behavior during the ENGAGE task the same between the beginning and the end of the sessions.

This is an important point to verify in interpretation of any fiber photometry data. As requested, we addressed the signal quality and the representation of behavioral events within the signal between the beginning of a session and the end of a session, for both sleep and the ENGAGE task (Figure R8). Despite the long (~10h) sleep recording sessions, our intermittent data acquisition resulted in stable signals (Figure S1C,D – also embedded here below) without any indication of significant photobleaching or phototoxicity affecting the results during these recordings.

Figure R8: No evidence for photobleaching during prolonged recordings in either the ENGAGE task or the sensory-evoked awakening (SEA) experiments. (left) Lick-evoked responses are compared between the first third and final third of trials, with no significant difference in signal quality. The interesting dip in the signal at 2-3 seconds following the lick that develops in the final third of the session may have to do with reward evaluation, and may be of interest for future investigation. (right) ACCp baseline levels immediately prior to auditory trials during NREM sleep. Gaps reflect intermittent periods when LEDs were shut off and no photometry signals were acquired for 30 minutes every 90 minutes (Methods). Note that ACCp activity remained globally stable across hours of recording (as in Figure S1D; see also Figure S1C).

Minor comments:

#21: - Fig2G legend mentioned $n = 20$ mice but I count 19 from the suppl table. Please clarify.

As written in the legend, single hemispheres were recorded from 18 mice, and 2 hemispheres were recorded in separate instances from one mouse. We have clarified this in the revised figure legend.

#22: - Lines 556-557: you wrote “To this end, the 465nm LED intensity was increased gradually until robust GCaMP/jRGECO fluctuations were observed above noise.” Are you using the same wavelength for RGeco as well or is it a typo? Was the power chosen arbitrarily or was a specific criterion used?

We thank the reviewer for capturing the typo. We did not use the same wavelength for jrGeco as it is excited by longer wavelengths than GCaMP. The criteria for choosing power are described in the methods section: ‘LED intensities were modulated for each recording site to allow the recording of viable signals with the minimal intensity. To this end, each LED intensity was increased gradually until robust GCaMP/jRGECO fluctuations were observed above noise. Next, the 405nm (isosbestic control channel) LED intensity was set to allow detection of motion artifacts. The total power at the tip of the patch cable was usually 0.05-0.1mW with slight individual variability.’

3- Claustrum activity and sleep

I found the relationship between calcium activity and sleep/wake state intriguing but also confusing and correlative. On top of the questions raised in the previous two sections, additional analyses and experiments are needed for this section in order to validate some of the conclusions. See specific comments below:

#23: - The link between depth of sleep, cortical responsiveness and claustrum activity is a bit confusing to me. Are we not expecting that claustrum activity should be stronger during REM than during non-REM sleep since REM sleep should be a light sleep state where it should be easier to wake up the mouse? Have the authors observed any relationship between sleep maintenance and photometry during REM?

The relation between claustrum activity and REM sleep is indeed intriguing, yet much more difficult to determine. As REM sleep is substantially rarer than NREM sleep in mice (8% of total time on average), and sounds were presented every 60sec, we could not accumulate enough trials of sensory-evoked awakening during REM sleep to allow a systematic analysis. To overcome this challenge, some studies perform selective REM sleep deprivation to induce more REM sleep (as in (Renouard et al., 2015)). However, this strategy introduces other confounds. Since REM sleep was not our focus, we did not employ this approach. Thus, to improve the clarity and focus in the revised manuscript and minimize complex interpretation, we removed analysis of REM sleep data from the revised manuscript.

#24: - The correlation between the depth of NREM sleep and fiber photometry is not really convincing. Though I can acknowledge that a correlation might exist, additional analyses and/or experiments must be performed to secure this claim. For example, the relationship plotted in Fig. 1G is extremely variable and some mice are plotted multiple times. The authors should show which data points are coming from the same animal. If they would average the data between animals (n=6) and not between experiments, would the relationship still be significant? Have the authors observed any effect of photobleaching? Is the difference seen in Fig 1G observed at different periods of the experiments (early v. late recording periods)?

We thank the reviewer for this important comment, which prompted us to present stronger data to support our conclusion. We have thoroughly revised the data addressing the relationship of ACCp activity to sleep (Figure 1 & Figure S1). Most importantly, we have performed a new experiment, in which we address the causal impact of elevating ACCp activity (with hM3Dq-CNO) on NREM sleep and sensory-evoked awakening (Figure 1F-L). The results of this causal experiment support and strengthen the original conclusions. In addition, we have also plotted the paired relationship of ACCp activity to awakening probability, as requested by the referee (revised Figure 1E). We have also plotted the signal over the time course of the experiments, in order to address possible changes in signal dynamics over prolonged behavioral sessions (Figures S1C,D). The new data and analyses demonstrate a causal relation between elevated ACCp activity and NREM sleep, as well as resilience to sensory-evoked awakening (Figures 1 & S1). The requested plots of the paired measurements within individual mice improve the clarity of the presentation (Figures 1C,E,I,L & S1E,G,I,J), while analysis of the signal over the course of prolonged sessions demonstrates the stability of the recordings (Figures S1C,D).

#25: - This part of the study is unfortunately only correlative. In order to be causal, reduce the confusion mentioned above and strengthen their claims, the authors must perform chemogenetic manipulations of ACCp and OFCp neurons (independently) and evaluate the consequences of altered claustrum activity on sleep patterns and on the number of auditory awakening. The authors might consider testing either inhibitory or excitatory DREADDS, if possible.

We have invested great effort in performing the requested experiment, expressing the Gq-DREADD hM3Dq in ACCp neurons, as well as GFP in control mice, following which we obtained undisturbed sleep recordings and sensory-evoked arousal recordings (double-blinded) from 6 ACCp vs 6 GFP mice (Figure 1H-L & Figure S1H-J). This experiment provided a clear result of increased time spent in NREM as well as a reduced probability to awaken following tone presentation in CNO-treated ACCp-Gq mice, but not GFP controls.

Minor comments:

#26: - Fig. 1 and Fig S2A: For consistency, use either “NREM maintained” or “Sleep maintained”

The labels of both figures describe the comparison of ‘maintained’ to ‘awakening’.

#27: - Line 107-108: “Operationalizing NREM sleep depth via the ratio between SWA and Theta, we observed that it shows a positive linear relationship with ACCp activity”. Is the relationship absent in OFCp activity?

As mentioned in response to the other reviewers, we have now removed the emphasis on SWA from the manuscript, explicitly focusing on the relationship between ACCp activity and NREM sleep to avoid confusion. As such, we have removed the SWA/theta metric from the revised version of the manuscript. However, as can be seen in Figure S1E-G, in contrast to ACCp neurons, the signal from OFCp neurons does not exhibit a relation of activity to NREM, nor a relation to maintained sleep.

4- Engage task and claustrum activity

#28: *The authors developed a novel task that they called ENGAGE. Though the task is interesting, it was hard evaluating it (data are distributed everywhere in the manuscript and description of the parameters is rather limited). It is problematic since the task has never been described nor validated before. As an example, I found difficult assessing the claims made by the authors regarding the interpretation of what the task is testing. The authors wrote several times the ENGAGE task is cognitively demanding, requiring impulse control, sustained and selective attention..., but what are the data supporting some of these statements. I found the task challenging in terms of sensory perception/discrimination for sure but I have hard time estimating how cognitively demanding the task is. Most trials require discriminating between different auditory cues, signaling different information such as trial onset, go signal and the cloud. In my opinion, the task challenges different features of auditory perception: 1- discrimination between trial onset cue and go cue, 2- hearing threshold (when the go cue intensity is varied) and 3- background/foreground discrimination when the cloud is present (similar to a cocktail party problem; I don't know if I would call the cloud a distractor).*

Based upon this comment we have drastically modified the text, better structuring the description of the ENGAGE task, and benchmarking our interpretation against the literature. We have also elaborated on the possible interpretation of results as a function of cognitive demand vs auditory perception.

#29: *It is also difficult comparing the freely moving data to the head-fixed data since the task rules are not comparable (for example, what is the penalty for impulsive trials when mice are head-restrained? plain water was used in freely moving sessions and sucrose was used in head-fixed sessions...). It is also not clear to me why the impulse rate doesn't change during the course of learning but decreases on the head-fixed rig (Fig. S3).*

The authors should spend more time describing the task and the different parameters. Right now, it is hard for the reader agreeing with the authors' claims. Alternatively, The authors should consider doing

a separate study describing the characterization of the task and perform additional controls (manipulate attentional levels, use different types of distractors and/or sensory cues such as air puffs, check for ACC/AUD contribution to the behavior?...).

We thank the reviewer for this comment, as it clarifies that inclusion of the description of behavior during training appears to have been distracting. Comparing the behavior of the mice while freely-moving vs. head-fixed is largely irrelevant, since mice were freely-behaving only during the training, and this behavior is not associated with measurements of claustrum activity or performance while head-fixed. We have removed the training data from the revised version of the manuscript, and will consider the reviewer's suggestion regarding a separate study aimed at characterizing different parameters within the task.

I have other specific comments:

#30: - The authors claim that ACCp neurons restricts engagement but could we imagine an alternative interpretation? The authors should precisely define their definition of engagement. To me, manipulating the ACCp neuron firing with the excitatory DREADD didn't change how the mice were engaged in the task since neither the total number of trials per sessions (Fig 3E), nor success rate (Fig S7C), nor reaction time (Fig S7D) were different between saline and CNO conditions. Behavior differed regarding the temporal distribution of the different trial types during the sessions. Fig 3E shows the cumulative distribution of various trial types during the course of 3 sessions (are the distributions for Impul/Hit/Miss plotted on top of each other? If it is the case, I would suggest plotting the data using stacking histograms). CNO reduced the number of impulsive trials and concomitantly increased the hit rate at the beginning of the sessions whereas more misses are observed at the end of each sessions. How can the authors disentangle a change in engagement, from a change in attention or a change in sensory perception? If hearing threshold, auditory discrimination and/or background/foreground discrimination are improved by the DREADD, then the main effect would be an improved sensory perception and not change in cognition or engagement. Naturally, since the mice are more quickly doing many hit trials, they are more rapidly satiated (not thirsty anymore or saturated by the sucrose consumption; by the way, can the authors explain why they used sucrose instead of plain water during the head-fixed behavior -Line 147- ?) and then they wouldn't be engaged in the task anymore and therefore do mostly missed at the end of the session.

We will relate to this comment, point by point:

- Mice performed the same number of trials: yes, the trials initiate automatically every 20 seconds, and sessions continue for 360 trials.
- Success rate and reaction time are not affected, on average, by CNO, and the major phenotype is an increase in impulse errors early in the trial, as well as an increase in misses late in the trial – indeed these are the observations. We find it hard to envision how this effect could be attributed to a change in sensory perception (threshold or discrimination). The mice either do or do not respond to the BBN 'no go' cue at trial onset. If the reduced impulsive responses are due to a reduction in perception – how could this be consistent with an increase in hit rate? The tones are not played concurrently – the BBN terminates at least 0.5 seconds prior to the 'go' cue – so we find it hard to explain the observations by a shift in sensory perception, auditory discrimination or background/foreground discrimination.
- Furthermore, plotting the psychometric curves describing behavior during the task, comparing performance in control (saline injections) vs CNO, early vs late in the session are inconsistent with a change in perception, as the results largely illustrate a shift in the impulse errors, with no major effect on the psychometric response curves (Figure R9, below).
- Regarding the possibility that a change in attention may underly the observation – we agree. Attention is a broad ('umbrella') term, and we perceive task engagement as belonging to the same category of cognitive functions as attention. However, we find engagement to be a more

useful term in the context of our observations, as a state of hyper-engagement is a substrate for impulse errors, while a state of hypo-engagement is a substrate for missed trials. The introduction of the revised manuscripts includes a definition of engagement: "Engagement describes the degree to which a subject is likely to interact with external sensory stimuli. Engagement exists along a continuum; from low levels of engagement in sleep, environmental detachment, and suppression of sensory-evoked responses, through varying levels during wakefulness. During wakefulness, states of disengagement can appear as sleep-like states and are accompanied by behavioral lapses, while states of hyper-engagement risk the execution of impulsive responses to irrelevant stimuli"

- We cannot rule out the possibility that satiety is contributing to the increase in misses at the end of the session. Indeed, the reason that the mice were rewarded with sucrose, rather than plain water, was in order to encourage participation, adding hedonic value beyond the simple recovery from thirst.

Figure R9: Comparison of psychometric behavior of hM3Dq mice in the presence or absence of CNO, early vs late in the session, illustrates no apparent deficit in sensory sensitivity.

- We have tried plotting the data as stacked histograms, as suggested by the reviewer, but our impression was that this did not improve the presentation of the data (Figure R10, below). We therefore retained the previous presentation (overlapping histograms). We did, however, include stacked bar graphs summarizing total trial outcome ratios over days in the experiment (Figure 3D).

Figure R10: Stacked histograms of trial outcome of hM3Dq mice, comparing CNO to saline.

#31: - The differences between ACCp and OFCp activity is questionable and could simply reflect a difference in sample size. For example, Fig. 2I and Fig S6B are not very different to my eyes. Doubling the sample size for the OFCp might lead to a similar significant difference. Furthermore, OFCp experiments used GCAMP6 indicator whereas ACC used primarily RGECCO indicator. Since the sensitivity and signal to noise ratio of the two indicators might differ substantially, how confident the authors are that the differences are not due to the use of two indicators. Can the authors plot the paired points for each animal?

As the reviewer requests, we have added the paired points (impulse error :: omission) for each animal. The data is based on 20 ACCp mice and 10 OFCp mice, sample sizes that are equivalent or greater than virtually any study we are aware of that performs an invasive investigation of the association of neural signal to behavior. This is definitely true for any study of the claustrum, the majority of which were based on at most n=6. For example, the undoubtedly elegant and influential study of (Fodoulouian et al., 2020) was based on n=4 or n=5 mice for every data point. Furthermore, our bootstrapping analysis evaluates the distribution of the data and extrapolates from it to account for sample size.

#32: - The chemogenetic effect is interesting but requires additional controls to convince the reader. First, I find the CNO concentration used, 10mg/kg, quite high. At such concentration, many side effects have already been reported in the literature (see for examples: 10.1016/j.neulet.2020.135432; 10.1101/2022.02.01.478652; 10.3389/fphys.2019.00521). It might be a concern since only 3 control mice were analyzed by the authors and they show the same tendency of reduced impulse rate. It would be important to do the same number of control mice and plot the data side by side with AACp. It's also very hard to compare the two groups since the authors used different normalization processes in Fig 3D and Fig S7B.

As requested, we have added additional control experiments, increasing the number of control mice to n=5, and have calculated and plotted the data for the control and experimental mice in a similar fashion (Figure 3 & Figure S5).

#33: - Considering all the technical limitations and questions raised up to now, I am not sure it makes too much sense discussing the Exploiters/Explorers dichotomy in great details. However, such difference might again represent individual differences in auditory perception. How can the authors test for that?

Furthermore, have the authors used the same splitting criterion for the OFCp mice? Are OFCp explorers showing a difference?

As all mice are littermates, co-housed and co-trained on the task, we have little reason to expect vast perceptual differences between them. However, in order to rule this possibility out, we will walk through the parameters we extracted for each mouse, based upon which we develop our interpretation of whether the mice are different in their perception or in their behavioral strategy. These are: a) psychometric curves of behavioral responses; b) Probability to perform impulse errors; c) Reaction times to the ‘go’ & ‘no go’ cue; d) Representation of sensory events in the photometry signal; e) Impact of tone cloud on psychometric curves and impulse errors; f) Distribution of ‘omission streaks’.

Looking at the psychometric curve of *exploiter* mice, one could hypothesize that they are deficient in auditory perception, as they appear to be less responsive to weak tones in the absence of a visual aid. However, *exploiter* mice also exhibited reduced sensitivity to the ‘no go’ BBN played on trial onset, as well as the tone cloud, both of which were delivered at the same intensity as the prominent ‘go’ cues. The specificity of the sensory sensitivity, retaining selective responses to prominent auditory tones at 6Khz, while ignoring other tones played at the same intensity, is inconsistent with a sensory deficit. Furthermore, one might expect these mice to respond, on average, more slowly to the auditory cues, which is not the case.

We find it more likely that these mice developed an attentional strategy in which they selectively respond to the visual cue and prominent ‘go’ cues, while forgoing the low intensity ‘go’ cues. This hypothesis is backed by: a) The psychometric curves of these mice, exhibiting selectivity to the visual aid and prominent auditory tones; b) The reduced impulse errors of these mice; c) Decoupling of their impulse errors from the timing of the ‘no go’ BBN; d) *Exploiter* mice exhibit a unique representation of the visual aid and prominent ‘go’ cues in their ACCp signal, lacking a representation of the ‘no go’ BBN and weak ‘go’ cues, to which they are not responsive; e) *Exploiter* mice are insensitive to the tone cloud mask, both in their psychometric curves, as well as in their impulse error rates; f) *Exploiter* mice exhibit shorter streaks of consecutive omissions, suggesting that their engagement in the task is more continuous throughout the session, and less susceptible to fluctuations than that of *explorer* mice.

Regarding OFCp mice, they were already included in the panel, and some of them were co-recorded with ACCp channels. We find that they largely fall within the ‘*exploiter*’ group (Figure R11). As there are only 2/10 mice that fall within the ‘*explorer*’ group, we cannot perform a meaningful comparison. Furthermore, we would like to clarify again that the OFCp data serves as a point of contrast throughout the study, and the emphasis of the study is on the ACCp population of claustrum neurons.

Figure R11: Distribution of OFCp with regard to task strategy. OFCp mice are circled, in the plot as presented in Figure 4A.

#34: - What would be the consequences of an inhibitory dREADD activation during the engage task? Would the mice only do impulsive trials?

We have not performed this experiment. However, in previous work from our lab (Atlan et al., 2018), we found that inhibiting EGR2-expressing claustrum neurons with Gi DREADDs resulted in an increase in impulsive errors. As described above, the population of EGR2⁺-claustral neurons likely overlaps with the population of claustral neurons captured by their projection to the ACC. We refer to this result in the discussion.

Minor comments:

#35: - *Since both GCaMP6s and RGECO was used, I find Fig 2D drawing misleading. It should be updated.*

The use of both indicators is now mentioned in the legend for this figure. In addition, separate schemes detailing the viral approach for dual recordings appear in Figure S2.

#36: - *Line 534 “These blocks were repeated 2-4 times (as long as mice maintained participation)” how was participation evaluated? Criteria?*

Blocks were designed to last 40 minutes, comprising 120 trials. Barring technical errors, each block was allowed to run in its entirety to maintain a fixed ratio of all randomized trial conditions. A new block was not initiated if during a block (typically towards its end), participation was low. However, there was no hard criterion for this. Barring a few exceptions in which mice kept participating, the maximum number of blocks was 4. Throughout the DREADD experiments, the block number was set to 3. We have related to this in the revised methods section of the manuscript.

ENGAGE task

#37: -*How was the water restriction done?*

Mice received water only as a reward for performance in the task. Per regulations of our IACUC protocol, if mice did not participate enough to receive at least 1ml of water, they were supplemented with water to reach that limit, although this was very rare, as mice were already trained in the task. Mice were weighed daily to ensure their body weight did not drop below 80%. We have related to this in the revised methods section of the manuscript.

#38: -*Is the re-initiation (leaving the port) also needed for misses?*

Yes. However, please note that this is true only for the automated training and irrelevant to the experiments described in the manuscript. On the rig, trials automatically initiated every 20 seconds.

#39: - *Since it is a novel task, an image of the apparatus might be helpful to add for the reader*

As suggested by the reviewer, we will devote a separate study to describing the training paradigm. The training apparatus is similar to the device we used for other automated behavioral analyses (Peretz-Rivlin et al., 2024; Terem et al., 2023).

#40: - *Do the mice need to reach 50-70% success before advancing to the next stage?*

In principle yes. However, this relates to the training data, which is de-emphasized in the revised version of the manuscript, and to which we will devote a separate manuscript (in development).

#41: - *Sucrose concentration and drop size should be indicated in the methods*

We have included the requested information in the revised methods section. Reward was a 4ul drop of 10% sucrose water.

Reviewer #3 (Remarks to the Author):

This study by Atlan and collaborators investigated the role of claustrum cells in sensory (dis)engagement in a novel cognitive task. Parallel data showing opposite correlation between sleep slow wave activities and ACC-projecting cell activity in the claustrum is proposed as a mechanism for lapses and missed trials in the cognitive task.

Both aspects of the work are very interesting and supported by studies in animal model and human showing (1) the role of the claustrum in cognitive task as well as sleep oscillations coordination across the cortex; and, (2) the importance of proper arousal, and the negative impact of slow waves occurrence during wakefulness (e.g., high sleep pressure), on cognitive performance. These lines of research have strong relevance to ageing and dementia pathologies.

We thank the reviewer for their supportive comments.

#1: Beside the writing of some of the results section that are difficult to follow but could be addressed in revision, I have several concerns on the experimental strategy and the significance of the cell dynamics reported here which would need additional data to supported the claims of the authors. First, the link between sleep SWA and disengagement is only assessed indirectly which are inconsistent with the major goal of the study (see rationale and goal in L22-28 and introduction section) and are not linked in the conclusions. This study reported changing activities of ACCp cells across sleep states, and cognitive task in separate experiments which only provide indirect link but left the untouched the original hypothesis. The latter would require to record SWA activities and ACCp cell dynamics during the cognitive task. Without that the title and the conclusions of the study are not supported since they may be the results of a change in ACCp cell activity that is not be associated with SWA. In other words, the implicit questions from this study remain open: Do SWA correlate with disengagement? Is the claustrum specifically implicated in this? the authors provide manipulation of claustral cell, but do those perturbations reflect awake SWA ? Potentially one may claim that any circuits in the cortex may have the same effect ? Thus, it is not clear what is specific about the claustrum here as compared to another cortical circuits. Accordingly, PFC and other input circuits are also involved in similar cognitive task... If the goal of the study was limited to the cognitive task, without the assessment of SWA modulation of claustrum cell dynamics, then this should be clarified and the manuscript should be re-written and the data from figure 1 removed.

We thank the reviewer for this important comment. Rereading the introduction section of the manuscript, we can see how a reader might develop an expectation that the study will focus on associating claustrum activity, SWA and task performance. More generally, following this and other comments and questions raised also by the other referees, we realized that the revised manuscript would benefit from focusing on the relationship between ACCp activity and behavioral engagement (i.e. the propensity to interact with external stimuli across different behavioral states). We have therefore revised the manuscript, removing the emphasis on SWA, and as suggested by the reviewer, removed the SWA data from figure 1. The study is now more clearly framed around the association of ACCp activity with behavioral engagement, both during sleep and its maintenance in the face of auditory stimulation (Figures 1, S1), and extending this logic to ACCp activity and task engagement during behavior (Figures 2-4, S2-S6). SWA is referred to in the discussion, where we speculate on a possible expanded mechanistic explanation for our findings.

My second concern relate to some of the figure panels and related experimental procedures, as listed below, where the difference of activities of claustrum cell dynamics appear minimal as reported (Z-scoring value are mainly below 0.8). In this reviewer's view, it is difficult to assessed what such small changes means without having access to the raw $\Delta F/F$ values.

#2: -Fig 1G: z-score values are quite low – how significant are those changes ? the authors must report $\Delta F/F$ values here.

Z-scoring the data was essential to be able to compare data between animals, as signal intensity varied between mice due to the position of the fiber relative to infected cells and the number of cells labelled under the fiber. However, the dynamics of these signals were consistent using both analyses. We are confident that z-scoring had little effect on the data, and attach an example of averaged data from a single mouse before and after z-scoring to demonstrate this (Figure R12). See also the response to comment #6, below.

Regarding the magnitude of claustrum dynamics, it again should be noted that the data was globally z-scored across the full recording sessions. Thus, the variance at all time scales across the whole session contribute to defining the magnitude of individual events.

Figure R12: Comparison of the average pre-tone ACCp activity prior to awakening trials (red trace) or maintained sleep (gray trace) for z-scored dF/F data (left) and raw dF/F (right), for mouse #5 that was included in the sensory-evoked awakening experiment. Shaded area represents s.e.m.

#3: *The sensory disengagement and infrequent sensory-evoked awakenings is not obvious from the results shown in the figure 1. The author should clarify and justify why some animal did not follow the same number of sessions (9 of 11 for some); how many animals ? how are data analyzed? From the results presented in figure 1, the authors claimed that “These results indicate that ACCp activity mirrors sleep depth and reduced sensory engagement”. This statement is correlative at best, and the evidence on sensory disengagement is weak. Simultaneous experimental measures of SWA are necessary to support this claim (see comment above and below).*

As requested by the reviewer, we have considerably revised this figure and the presentation of the data. Data is now presented for individual mice instead of the recording sessions (Figures 1C,E,I,L & S1E,G,I,J). We have revised the methods section as well to clarify the analysis performed. Also, we have removed the SWA data.

In order to go beyond correlative statements, we have added a completely new causal experiment, in which we address the impact of Gq-DREADD activation of ACCp neurons on NREM sleep and tone-evoked awakening (Figure 1H-L; Figure S2H-J).

#4: *-Fig 2: if I understand well, the EEG/EMG were not recorded during the experiment, so how is this assessing SWA ?*

We did not assess SWA in the context of the ENGAGE task, and indeed there was no electrophysiology recordings of EEG/EMG during the task. The sleep experiments with EEG/EMG were performed in the Nir lab at Tel-Aviv University, while the experiments in the context of the ENGAGE task were performed in the Citri lab at the Hebrew U. (Jerusalem). The manuscript has been extensively revised, removing the relation to SWA from the introduction and results, alleviating the potential for confusion around this point.

#5: *My understanding is that all trials from 20 mice are pooled together. Thus the high value of trials (>25000) is misleading and due to only few animals. Although this is informative I would suggest to separate results for each animals instead of pooling all datasets. In addition, there are as many hit as*

missed trials? Was this expected? If animals learned the task, one would expect a decrease of miss, isn't it?

The large number of mice actually contributed equally to the results, since sessions typically extended for an equal number of trials (commonly 360 trials). All statistics were calculated taking individual mice into account as a variable.

Regarding the high proportion of errors – this was by design. Chance level performance in the task would be near 0%, as mice must learn to refrain from licking during the delay period for the target Go cue to be presented, but lick when it is presented. This added complexity of the task was explicitly designed in order to retain a high level of errors of both types (impulsive errors and misses), as the emphasis of the analysis is on claustrum signal associated with task failures. Had the mice maintained >80% success during recording, it would have been difficult to assay the association of claustrum activity to impulse or omission errors. Indeed, during some stages of training, mice achieved, on average, above 80% hit rate, illustrating a solid understanding of the task, which was challenged in the full task, following addition of all task variables.

#6: -Fig 2E-I: Z-scores of data are quite low suggesting minimal differences between experimental conditions. Please present raw data in parallel and explain the relevance of such low Z-scores.

The z-scores are low as they are calculated over the raw signal from the entire session. However, the effects are robust, as evident in the figures. As mentioned above with respect to the sleep data analysis, z-scoring was imperative for between-animal comparisons. The raw values of fiber photometry depend on multiple parameters such as the distance between the fiber and the indicator-expressing neurons, as well as the density of the neurons and the power settings of the source LED. As such, the absolute value of the signal varies between animals, as does the magnitude of changes ($\Delta F/F$). The single trial data in Figure 2E-G provides a reference for the prominence of individual events. In figure R13 we provide a reference for the impact (or lack thereof) of z-scoring on the average signal, as well as the utility of z-scoring in enabling comparisons across mice.

#7: Regarding the task itself, Why tone-cloud distractor and visual stimuli are presented in some of the trials and not all of them? It is unclear how these are affecting performance as data do not systematically take into account these 'co-occurring' stimuli. Do they have any effect on the performance ? or ACCp activation, etc? please clarify.

We thank the reviewer for this comment, clarifying that our description of the task was previously not optimal. We have extensively revised the writing in the hope that the current explanation is clearer. The different variables were included in subsets of the trials so as to enable us to identify the relative impact of each variable on performance (Figure 2C, Figures 4A,B,C), as well as to enable the dissociation of the representation of each variable in the signal of claustrum neurons (Figure 2E-G, Figure S3, Figure S4, Figure 4E). The tone-cloud distractor had significant effects on both impulsive errors and on the ability to detect low-intensity Go cues (Figure 2C). The degree of this effect was used to define two separate strategies in the task (Figure 4A). 'Exploiter' mice, which were less affected by the tone-cloud, utilized the visual aid to succeed in most trials in which it was presented (Figure 4B). While ACCp responses to the Go cue were higher when the visual stimulus was present (Figure 2G), this response occurred after trial outcome had been determined, and we found no relationship between ACCp responses to the Go cue or to the tone-cloud and performance. Rather, we found that ACCp activity preceding the trial was important in determining its outcome.

#8: -Fig 3B: Does CNO/DREADDS activation of ACCp reflect activity similar to SWA or sleep ? it seems to induce massive waves of activity with a very long periodicity (± 20 s ?) much slower than previously reported SWA – what does that mean ? Are those similar to natural/spontaneous activity ? The effect on the performance is clear, however, it is difficult to conclude that this is due to SWA. Note that proper control for CNO experiments is actually CNO injection in DREADD-negative animals since the compound itself may have effect on brain dynamics and behaviors.

Regarding the first question, we believe that the example trace we chose may have been a bit of an extreme case, creating an exaggerated impression of the effect of CNO on ACCp activity. We have therefore replaced this trace with one that better represents the magnitude of the CNO effect (Figure 3B). We have also included quantification of the effect of CNO on mean z-score, event rate, event width and event magnitude (Figure S5A). Across mice, the effect of CNO is of elevating overall activity and amplifying the size of events, with no effect on their periodicity.

Since, as the reviewer pointed out, we did not record EEG during the ENGAGE task, we could not directly probe SWA during behavior. However, the added DREADD experiment in our sleep studies establish that CNO/DREADDs increase the occurrence of NREM sleep. A plausible hypothesis would be that elevated ACCp claustrum activity during the task may also lead to sleep-like processes including SWA invading the activity during wakefulness, but this would need to be tested properly in the future.

Regarding the comment about CNO controls, we agree and have performed the CNO control as suggested, and included additional repetitions of this experiment in revision, as requested by the other reviewers (Figure 3C and Figure S5F).

#9: -L157: why were animals tested in a head-restrained set up while training was in freely-behaving context ? This may have major influence on performance, SWA etc. and I don't see how this is being controlled or tested for in the data set. Please clarify.

We understand that the inclusion of the training data was confusing and have therefore reduced its presence in the manuscript. Claustrum photometry and behavioral performance are only reported from head-fixed mice, not from freely moving mice.

We performed training in an automated homecage setup simply because this reduced the resources required in training, reduced load on the main recording rig, and increased the number of available mice as mice quickly transferred their training from the automated setup to the contingencies on the rig. We found that automated homecage training accelerated the performance of the ENGAGE task while head-fixed, achieving desired levels of performance after 2-3 days of habituation to head-fixation, in contrast to multiple (~5-10) weeks of training which are common practice for complex paradigms under head-fixation.

Minor concerns:

#10: L76: ACC-projecting neurons in the claustrum (ACCp) is confusing – the authors should use CIACC or other alternative names but this should be clarified.

As we have already used the term ACCp in previous published work (Terem et al., 2023), we prefer to maintain this terminology for consistency.

#11: L89: “reliance” – please reword.

OK. The text has been extensively revised.

#12: Fig1A: “R” form retro-AAV is missing

Corrected.

#13: Fig 1C: color coding of dots is difficult to see – please adapt.

OK. This panel has been adapted and moved to Figure S1A.

#14: Fig 1E: could the authors provide quantification and statistics for those those graph ?

Based on the reviewers' comments, we have shifted the focus of figure 1 to the relationship of ACCp activity with NREM & sensory-evoked awakening, and have removed former panel 1E from the manuscript.

References:

- Atlan, G., Matosevich, N., Peretz-Rivlin, N., Yvgi, I., Chen, E., Kleinman, T., Bleistein, N., Sheinbach, E., Groysman, M., Nir, Y., & Citri, A. (2021). Claustal Projections to Anterior Cingulate Cortex Modulate Engagement with the External World. *BioRxiv*, 2021.06.17.448649. <https://doi.org/10.1101/2021.06.17.448649>
- Atlan, G., Terem, A., Peretz-Rivlin, N., Sehrawat, K., Gonzales, B. J., Pozner, G., Tasaka, G. ichi, Goll, Y., Refaeli, R., Zviran, O., Lim, B. K., Groysman, M., Goshen, I., Mizrahi, A., Nelken, I., & Citri, A. (2018). The Claustrum Supports Resilience to Distraction. *Current Biology*, 28(17), 2752-2762.e7. <https://doi.org/10.1016/j.cub.2018.06.068>
- Chia, Z., Augustine, G. J., & Silberberg, G. (2020). Synaptic Connectivity between the Cortex and Claustrum Is Organized into Functional Modules. *Current Biology*, 30(14), 2777-2790.e4. <https://doi.org/10.1016/j.cub.2020.05.031>
- Erwin, S. R., Bristow, B. N., Sullivan, K. E., Kendrick, R. M., Marriott, B., Wang, L., Clements, J., Lemire, A. L., Jackson, J., & Cembrowski, M. S. (2021). Spatially patterned excitatory neuron subtypes and projections of the claustrum. *ELife*, 10. <https://doi.org/10.7554/ELIFE.68967>
- Fodoulian, L., Gschwend, O., Huber, C., Mutel, S., Leone, R., Renfer, J.-R., Ekundayo, K., Rodriguez, I., & Carleton, A. (2020). The claustrum-medial prefrontal cortex network controls attentional set-shifting. *BioRxiv*, 2020.10.14.339259. <https://doi.org/10.1101/2020.10.14.339259>
- Grienberger, C., Adelsberger, H., Stroh, A., Milos, R. I., Garaschuk, O., Schierloh, A., Nelken, I., & Konnerth, A. (2012). Sound-evoked network calcium transients in mouse auditory cortex in vivo. *Journal of Physiology*, 590(4), 899–918. <https://doi.org/10.1113/JPHYSIOL.2011.222513>
- Jackson, J., Karnani, M. M., Zelman, B. V., Burdakov, D., & Lee, A. K. (2018). Inhibitory Control of Prefrontal Cortex by the Claustrum. *Neuron*, 99(5), 1029-1039.e4. <https://doi.org/10.1016/j.neuron.2018.07.031>
- Legaria, A. A., Matikainen-Ankney, B. A., Yang, B., Ahanonu, B., Licholai, J. A., Parker, J. G., & Kravitz, A. V. (2022). Fiber photometry in striatum reflects primarily nonsomatic changes in calcium. *Nature Neuroscience* 2022 25:9, 25(9), 1124–1128. <https://doi.org/10.1038/s41593-022-01152-z>
- Lipton, D. M., Tamimi, M., Shalom, I., Sheinfeld, T., Gonzales, B. J., Groysman, M., & Citri, A. (2023). Striatal calcium transients detected by fiber photometry propagate to axons. *BioRxiv*, 2023.10.09.560813. <https://doi.org/10.1101/2023.10.09.560813>
- Marriott, B. A., Do, A. D., Zahacy, R., & Jackson, J. (2021). Topographic gradients define the projection patterns of the claustrum core and shell in mice. *The Journal of Comparative Neurology*, 529(7), 1607–1627. <https://doi.org/10.1002/CNE.25043>
- Narikiyo, K., Mizuguchi, R., Ajima, A., Shiozaki, M., Hamanaka, H., Johansen, J. P., Mori, K., & Yoshihara, Y. (2020). The claustrum coordinates cortical slow-wave activity. *Nature Neuroscience*, 23(6), 741–753. <https://doi.org/10.1038/s41593-020-0625-7>
- Ntamati, N. R., Acuña, M. A., & Nevian, T. (2023). Pain-induced adaptations in the claustrum-cingulate pathway. *Cell Reports*, 42(5). <https://doi.org/10.1016/J.CELREP.2023.112506>
- Peng, H., Xie, P., Liu, L., Kuang, X., Wang, Y., Qu, L., Gong, H., Jiang, S., Li, A., Ruan, Z., Ding, L., Yao, Z., Chen, C., Chen, M., Daigle, T. L., Dalley, R., Ding, Z., Duan, Y., Feiner, A., ... Zeng, H. (2021). Morphological diversity of single neurons in molecularly defined cell types. *Nature*, 598(7879), 174–181. <https://doi.org/10.1038/S41586-021-03941-1>
- Peretz-Rivlin, N., Marsh-Yvgi, I., Fatal, Y., Terem, A., Turm, H., Shaham, Y., & Citri, A. (2024). An automated group-housed oral fentanyl self-administration method in mice. *Psychopharmacology*. <https://doi.org/10.1007/S00213-024-06528-6>

- Qadir, H., Krimmel, S. R., Mu, C., Pouloupoulos, A., Seminowicz, D. A., & Mathur, B. N. (2018). Structural Connectivity of the Anterior Cingulate Cortex, Claustrum, and the Anterior Insula of the Mouse. *Frontiers in Neuroanatomy*, *12*. <https://doi.org/10.3389/FNANA.2018.00100>
- Renouard, L., Billwiller, F., Ogawa, K., Clément, O., Camargo, N., Abdelkarim, M., Gay, N., Scoté-Blachon, C., Touré, R., Libourel, P. A., Ravassard, P., Salvert, D., Peyron, C., Claustrat, B., Léger, L., Salin, P., Malleret, G., Fort, P., & Luppi, P. H. (2015). The supramammillary nucleus and the claustrum activate the cortex during REM sleep. *Science Advances*, *1*(3). <https://doi.org/10.1126/SCIADV.1400177>
- Reser, D. H., Majka, P., Snell, S., Chan, J. M. H., Watkins, K., Worthy, K., Quiroga, M. D. M., & Rosa, M. G. P. (2017). Topography of claustrum and insula projections to medial prefrontal and anterior cingulate cortices of the common marmoset (*Callithrix jacchus*). *Journal of Comparative Neurology*, *525*(6), 1421–1441. <https://doi.org/10.1002/cne.24009>
- Smith, J. B., Alloway, K. D., Hof, P. R., Orman, R., Reser, D. H., Watakabe, A., & Watson, G. D. R. (2019). The relationship between the claustrum and endopiriform nucleus: A perspective towards consensus on cross-species homology. *Journal of Comparative Neurology*, *527*(2), 476–499. <https://doi.org/10.1002/cne.24537>
- Terem, A., Fatal, Y., Peretz-Rivlin, N., Turm, H., Koren, S. S., Kitsberg, D., Ashwal-Fluss, R., Mukherjee, D., Habib, N., & Citri, A. (2023). Claustral neurons projecting to frontal cortex restrict opioid consumption. *Current Biology*, *33*(13), 2761–2773.e8. <https://doi.org/10.1016/j.cub.2023.05.065>
- Terem, A., Gonzales, B. J., Peretz-Rivlin, N., Ashwal-Fluss, R., Bleistein, N., del Mar Reus-Garcia, M., Mukherjee, D., Groysman, M., & Citri, A. (2020). Claustral Neurons Projecting to Frontal Cortex Mediate Contextual Association of Reward. *Current Biology*, *30*(18), 3522–3532.e6. <https://doi.org/10.1016/j.cub.2020.06.064>
- Tischbirek, C. H., Birkner, A., & Konnerth, A. (2017). In vivo deep two-photon imaging of neural circuits with the fluorescent Ca²⁺ indicator Cal-590. *Journal of Physiology*, *595*(10), 3097–3105. <https://doi.org/10.1113/JP272790>
- Wang, Q., Wang, Y., Kuo, H. C., Xie, P., Kuang, X., Hirokawa, K. E., Naeemi, M., Yao, S., Mallory, M., Ouellette, B., Lesnar, P., Li, Y., Ye, M., Chen, C., Xiong, W., Ahmadinia, L., El-Hifnawi, L., Cetin, A., Sorensen, S. A., ... Koch, C. (2023). Regional and cell-type-specific afferent and efferent projections of the mouse claustrum. *Cell Reports*, *42*(2). <https://doi.org/10.1016/J.CELREP.2023.112118>
- White, M. G., Cody, P. A., Bubser, M., Wang, H.-D., Deutch, A. Y., & Mathur, B. N. (2017). Cortical hierarchy governs rat claustrorocortical circuit organization. *Journal of Comparative Neurology*, *525*(6), 1347–1362. <https://doi.org/10.1002/cne.23970>
- White, M. G., & Mathur, B. N. (2018a). Claustrum circuit components for top–down input processing and cortical broadcast. *Brain Structure and Function*, *223*(9), 3945–3958. <https://doi.org/10.1007/s00429-018-1731-0>
- White, M. G., & Mathur, B. N. (2018b). Frontal cortical control of posterior sensory and association cortices through the claustrum. *Brain Structure and Function*, *223*(6), 2999–3006. <https://doi.org/10.1007/s00429-018-1661-x>
- White, M. G., Panicker, M., Mu, C., Carter, A. M., Roberts, B. M., Dharmasri, P. A., & Mathur, B. N. (2018). Anterior Cingulate Cortex Input to the Claustrum Is Required for Top-Down Action Control. *Cell Reports*, *22*(1), 84–95. <https://doi.org/10.1016/j.celrep.2017.12.023>

REVIEWERS' COMMENTS

Reviewer #1 (Remarks to the Author):

The authors have greatly clarified their manuscript and improved the message significantly. This is a very interesting paper, and the main claim regarding the claustrum activity in this task is very well presented and discussed. I feel this paper should be published. However, I have one nagging question.

I have some concerns about the new experiments conducted with hm3d presented in Fig1 and SFig1. Here the authors used 10mg/kg CNO, which is a very high dose. They do the proper controls (SFig1H-J). However, there was never a statistical comparison between control and hm3d mice. 5 of the 7 mice in the control experiment show profound increases in NREM similar to hm3d mice. The amount of time in NREM is very similar in hm3d and control mice following CNO, but these groups have very different response to saline it appears. I would just like to see what the authors have to say about this, and to get their opinion on how confident they are regarding the claims made with this data. They could do this either in a response to review, or in the text, it is up to them.

Please correct the number of control mice used to $n = 7$ in the main text. Line 113

Reviewer #3 (Remarks to the Author):

I thank the authors for seriously addressing my comments and those from other reviewers.

NCOMMS-22-40947A-Z Response to Reviewer Feedback:

Reviewer #1:

The authors have greatly clarified their manuscript and improved the message significantly. This is a very interesting paper, and the main claim regarding the claustrum activity in this task is very well presented and discussed. I feel this paper should be published. However, I have one nagging question. I have some concerns about the new experiments conducted with hm3d presented in Fig1 and SFig1. Here the authors used 10mg/kg CNO, which is a very high dose. They do the proper controls (SFig1H-J). However, there was never a statistical comparison between control and hm3d mice. 5 of the 7 mice in the control experiment show profound increases in NREM similar to hm3d mice. The amount of time in NREM is very similar in hm3d and control mice following CNO, but these groups have very different response to saline it appears. I would just like to see what the authors have to say about this, and to get their opinion on how confident they are regarding the claims made with this data. They could do this either in a response to review, or in the text, it is up to them.

Please correct the number of control mice used to $n = 7$ in the main text. Line 113

We thank the reviewer for their comment. Thank you for catching the error regarding the n # of control mice. We have corrected this.

Regarding the comparison of control and hM3Dq-expressing mice, we concede that there is substantial variance in the behavior of the mice under control (saline) conditions. These are exceedingly difficult experiments, and sleep is obviously a highly variable metric between individuals. Nonetheless, we are confident enough regarding our claims to include them in the manuscript, and attach our names to them, for posterity. Our confidence is based on the following: 1) Performing a paired 2-tailed t-test, we do not observe a difference in the sleep between control and hM3Dq-expressing mice under the saline condition; 2) Comparing the effect of CNO between the control and hM3Dq-expressing group as a change in NREM sleep induced by CNO vs saline, we observe a statistically significant effect (paired 2-tailed t-test, $t(12) = -2.2$, $p < 0.05$). This result provides further that activation of hM3Dq in ACCp claustral neurons affects sleep beyond any baseline difference in saline, and beyond any potential off-target effect of CNO.